# Anionic polymerization of nonaromatic maleimide to achieve full-color nonconventional luminescence

Xin Ji[1,2], Weiguo Tian [1✉], Kunfeng Jin[1], Huailing Diao[1,2], Xin Huang[3], Guangjie Song [1✉] & Jun Zhang [1,2✉]

Nonconventional or nonconjugated luminophore without polycyclic aromatics or extended π-conjugation is a rising star in the area of luminescent materials. However, continuously tuning the emission color within a broad visible region via rational molecular design remains quite challenging because the mechanism of nonconventional luminescence is not fully understood. Herein, we present a new class of nonconventional luminophores, poly(maleimide)s (PMs), with full-color emission that can be finely regulated by anionic polymerization even at ambient temperature. Interestingly, the general characteristics of nonconventional luminescence, cluster-triggered emission, e.g., concentration-enhanced emission, are not observed in PMs. Instead, PMs have features similar to aggregation-caused quenching due to boosted intra/inter-molecular charge transfer. Such a biocompatible luminescent material synthesized from a low-cost monomer shows great prospects in large-scale production and applications, including security printing, fingerprint identification, metal ion recognition, etc. It also provides a new platform of rational molecular design to achieve full-color nonconventional luminescence without any aromatics.

[1] Beijing National Laboratory for Molecular Sciences, CAS Key Laboratory of Engineering Plastics, Institute of Chemistry Chinese Academy of Sciences (CAS), 100190 Beijing, China. [2] University of Chinese Academy of Sciences, 100049 Beijing, China. [3] School of Chemical Engineering and Technology, Tianjin University, 300072 Tianjin, China. ✉email: wgtian@iccas.ac.cn; gsong@iccas.ac.cn; jzhang@iccas.ac.cn

Since the first spark of fire, mankind has never stopped pursuing new luminescent materials, from the primitive long-afterglow phosphorescent minerals found in nature to cutting-edge artificial luminescent materials (quantum dots, carbon dots, perovskite, etc.)[1–4]. In recent years, the development of organic luminescent materials has grown rapidly due to their competitive advantages, such as incredibly diverse molecular structures, finely tunable emission, and high quantum yield compared to other artificial luminescent materials. Traditional organic luminophores can be generally categorized into aggregation-caused quenching (ACQ)[5,6] and aggregation-induced emission (AIE)[7,8] materials according to their different intrinsic emission characteristics. ACQ or AIE luminophores are always composed of extended π-electron or conjugation systems with wide-range electron delocalization (e.g., polycyclic or heterocyclic aromatics), which is the origin of their photoluminescence in the visible region (400–700 nm).

Nonconventional luminescence has a long history[9] and has just been revived in recent years. Different from ACQ and AIE luminogens, nonconventional luminophores contain only electron-rich heteroatoms with lone-pair electrons (e.g., N, O, S, and P) or isolated unsaturated bonds (e.g., C=O, C=C, C=N, C=O, and C≡N) that usually are not considered as the typical luminophores. Previously, only isolated cases of nonconventional luminescence were occasionally observed in some biomolecules and natural polymers, e.g., enzymes, proteins[10,11], starch, cellulose, and sodium alginate[12]. Inspired by previous pioneering studies, an increasing number of nonconventional polymeric luminophores that exhibit concentration-enhanced emission, excitation-dependent luminescence and even triplet emission have been discovered[13–22], e.g., polyethyleneimines, polyacrylonitrile, and poly(ethylene glycol)[23]. It should be noted that only saturated σ-bonds (–O–, –S–, –OH, –NH–, –NH$_2$, –SiO) or isolated unsaturated bonds (–COO–, –COOCO–, –NHCO–, –NHCS–, –OSO–, –PO$_3$–, –COOH, –CN, –C$_5$H$_5$N, –SO$_3$H) are involved in the backbone or pendent groups of the previous nonconventional luminescent oligomers, polymers, macromolecules, and supramolecules. Benefiting from these simple chemical structures compared with aromatic ACQ and AIE luminogens, nonconventional luminophores have many intrinsic merits that are desired in large-scale production and practical applications, e.g., short synthetic route, easy availability, and efficient processability. In particular, without potential cytotoxicity from cyclic aromatics, nonconventional luminophores are born with excellent biocompatibility, which is very attractive in biological and medical applications.

More and more new nonconventional luminophores are constantly being developed. However, the underlying mechanism of nonconventional luminescence is still under debate[17]. Many theoretical assumptions[13,21,24,25] have been proposed to reveal the underlying mechanism of nonconventional luminescence in the past. Unfortunately, there is no a general theory or model that has been finally concluded and commonly accepted. e.g., enhanced π–π stacking and other physical interactions (e.g., inter-, or intramolecular energy/charge transfer) for ACQ, restriction of intramolecular motion (RIM, rotation and vibration) for AIE. Due to the complexity and uncertainty of unconventional emission centers or species, most assumptions only touch the tip of the iceberg and even contradict each other. For example, the oxidation mechanism proposed to explain the fluorescence of PAMAM dendrimers[21] was completely overturned by the strong emission of siloxane-PAMAM without oxidation[16], which was attributed to the aggregation of carbonyl groups instead. To date, cluster-triggered emission (CTE)[26,27] seems to be the most prevailing system in nonconventional luminescence and its theory and mechanisms have been directly used to explain lots of

nonconventional luminescence phenomena without hesitation. For CTE luminogens, through-space conjugation (TSC) and rigidification of molecular conformation introduced by the clusterization of heteroatoms containing π-electrons, lone-pair electrons (e.g., –O–, –S–, –OH) or unsaturated bonds (e.g., C=O, C=C, C=N, and C≡N) are thought to be the origin of luminescence. Additionally, strong molecular interactions (e.g., hydrogen bond, ionic bond) that are beneficial to TSC and conformation rigidification can significantly enhance nonconventional luminescence. Therefore, concentration-enhanced emission like AIE and excitation-dependent luminescence (multiple luminescent centers) are generally observed in CTE. Nevertheless, the theory of CTE is still far from a satisfying universal mechanism of nonconventional luminescence, especially when rationalizing nonconventional luminescent systems without general characteristics of CTE[26,28]. Therefore, we need more typical nonconventional luminophores and reliable theoretical assumptions to perfect CTE and ultimately elucidate the underlying mechanism of nonconventional luminescence.

Continuous regulation of emission color through rational molecular design and enhanced emission efficiency comparable to that of traditional quantum dots (QDs) and AIEgens are considerable challenges hindering the practical application of nonconventional luminophores. Despite the excitation-dependent luminescent characteristics, the emission window of nonconventional luminophores is limited to 400–500 nm. Fortunately, a few pioneering works have brought hope for shifting the emission of nonconventional luminescence to the red and even NIR regions. For instance, polytriphenylmethanol powder with yellow photoluminescence originating from prolonged polymerization, orange–red fluorescent poly(itaconic anhydride-co-vinyl caprolactam) (PIVC) powder synthesized through free-radical copolymerization[29], red luminescent poly(maleic anhydride-alt-vinyl acetate) (PMV) powder prepared by heat treatment (180 °C, 5 min)[30], PMV with a fascinating emission color change from blue to red adjusted by changing the pH value[31] and aggregations of aliphatic cyclic imides (containing Br and I) with emission from blue to NIR[32] have been reported. Although all these nonconventional systems are claimed to follow the mechanism of CTE, the regulation principle remains elusive. Generally, there are three major principles to tune the emission of organic ACQ and AIE luminogens: the extension of π-conjugation[33], the introduction of through-bond intramolecular charge transfer by incorporation into donor–acceptor (D–A) units[34], and the facilitation of through-space charge transfer (TSCT) or intermolecular charge transfer[35] via spatially separated D–A groups on nonconventional polymer chains[36] or supramolecular self-assembly[37]. Notably, because of the unsettled mechanism of nonconventional luminescence, it is very difficult to achieve full-color emission of nonconventional luminophores by regular routines based on rational molecular design.

Maleimide and succinimide are the common building blocks seen in nonconventional luminescent small molecules[32,38–41] without any aromatics. The luminescent properties of these maleimide and succinimide derivatives fundamentally depend on the substituent groups containing different heteroatoms (e.g., –NH–, –S–, –Br, –I, etc.), intermolecular interactions, and aggregation/cluster. In addition, the luminescent nonconjugated polymers bearing succinimide side groups[42] or incorporated with maleimide units[43] have been discovered and their intrinsic emission can be controlled by regulating the aggregation of polymer chains. Maleimide is an important monomer. Polymaleimides (PMs) are a common commercial polymer used as a chelating agent, exchange resin and detergent and can be synthesized through free-radical polymerizations and anionic polymerization[44–47]. It should be noted that only small molecules

of maleimide derivatives and polymers with isolated maleimide moieties have been currently investigated and the luminescent PMs directly synthesized by polymerization of maleimide have not been reported yet. Herein, it is the first time that the intrinsic nonconventional full-color emission of PMs has been achieved by anionic polymerization of maleimide using mild Lewis bases as catalyst. The emission color of PMs can be continuously tuned across visible regions by changing polymerization conditions. In comparison with free-radical polymerization, anionic polymerization rearranges the connection of repeating units in PMs. A polymer chain with alternating –D–A– (electron donor and acceptor) electronic structure is generated, which can help establish the effective channels of intra-/intermolecular charge transfer, As a result, the emission colors of PMs are continuously shifted from blue to red. Interestingly, the criteria for CTE[48] cannot be mechanically applied to the nonconventional luminescence of PMs since general characteristics such as concentration-enhanced emission and excitation-dependent luminescence are not observed. Therefore, the full-color luminescent PMs reported herein are considered another exceptional example of clusteroluminescence or CTE. Such a fascinating full-color luminescent polymer or oligomer from a commercial monomer shows great prospect in the applications of security printing, fingerprint identification and metal ion recognition, more importantly, provides a new molecular platform to profoundly understand the intrinsic luminescence of nonconventional luminophores and to build nonconventional luminophores without aromatics.

## Results

### Synthesis and characterization of A-PM and Fr-PM.

Although PM is a commercial polymer from a common monomer, maleimide, its inherent blue and near-ultraviolet fluorescence (400–450 nm, Supplementary Fig. 1) has not received rigorous attention it deserves. As shown in Fig. 1a, mild Lewis bases (e.g., triethylamine, TEA) are employed to initiate the polymerization of maleimide instead of AIBN, and nonconjugated PMs (10-g scale in the lab, Supplementary Fig. 2) with full-color emission (red, green, and blue) can be easily achieved only by controlling polymerization conditions (Supplementary Table 5). The mass spectrometric fragment analysis of PMs synthesized via free-radical polymerization (Fr-PM, AIBN as initiator) and anionic polymerization (A-PM, TEA as initiator) presented in Fig. 1b clearly suggests that the molar masses of repeat units in Fr-PM and A-PM are the same (97 Da), exactly coinciding with the molecular weight of the maleimide monomer. In the FTIR spectra shown in Fig. 1c, Fr-PM has a broader stretching vibration peak ($1716 \, \text{cm}^{-1}$) of carbonyl groups, a more intense characteristic band ($3500–3000 \, \text{cm}^{-1}$) of N–H and a stronger vibration peak of –CH– ($2769 \, \text{cm}^{-1}$) than A-PM, which indicates that there are more N–H and –CH– bonds in Fr-PM. The characteristic peak ($2950 \, \text{cm}^{-1}$) of –CH$_2$– in A-PM is much more obvious than that in Fr-PM, which indicates that more –CH$_2$– structures exist in A-PM. The MS and FTIR results demonstrate that there are two different connection modes of repeating units in PMs, –C–C– and –C–N–, which are substantially influenced by polymerization methods without changing the molecular repeating unit. As the schematic polymer structures illustrate in Fig. 1a, the repeating units of Fr-PM are connected mainly by –C–C– units, therefore resulting in more N–H bonds. In contrast, –C–C– and –C–N– connections coexist in the polymer chain of A-PM, in which fewer N–H bonds and more –CH$_2$– units appear. This can be further confirmed by [1]H NMR and [13]C NMR DEPT analysis. In Fig. 1d, the N–H peak (11.6 ppm) of A-PM is remarkably lower than that of Fr-PM, indicating fewer N–H bonds in A-PM. In

addition, in comparison with those of Fr-PM, the [1]H NMR characteristics of A-PM in a high magnetic field less than 6 ppm are more complicated. The [1]H NMR spectrum of Fr-PM shows only the characteristic peaks of –CH– (2.5-4.5 ppm) ascribed to the repeating units in the –C–C– structure. However, in addition to the characteristic peaks (2.5–3.3 ppm) of –CH– in –C–C– units and –CH$_2$– in –C–N– units, a group of new peaks (4.5–5.5 ppm) that is obviously different from Fr-PM and attributed to –CH– in –C–N– units are observed in the [1]H NMR spectrum of A-PM. Such a large difference in structure between Fr-PM and A-PM can be further verified by [13]C NMR DEPT analysis. As shown in Fig. 1e, there are two distinct peaks that are ascribed to –CH– (47.1 ppm, 48.2 ppm) and –CH$_2$– (32.4 ppm, 33.4 ppm) in the spectrum of A-PM. However, only the characteristic peaks of –CH– (44.3 ppm, 43.7 ppm) exist in the spectrum of Fr-PM. Therefore, it can be reconfirmed that Fr-PM contains –C–C– connections and that A-PM is chemically linked by both –C–C– and –C–N–. Intrinsically, this polymerization-induced variation in the connection mode of repeating units should be the origin of the different fluorescent properties between Fr-PM and A-PM.

### Full-color emission of A-PM regulated by anionic polymerization.

Continuous precise tuning of fluorescent emission within the visible region or even in a larger range has always been the primary motivation for the structural design and synthesis of fluorescent materials. As concluded previously, the connection of repeating units changed by anionic polymerization should be responsible for the emission redshift of PMs. Hence, Lewis base species and reaction times related to anionic polymerization are investigated to regulate the emission of PMs. As shown in Fig. 2a and the Supplementary Fig. 6, the full-color fluorescent emission of A-PM powders or solutions (DMF as solvent) across the visible region (red–green–blue) is realized by using different Lewis bases (I, triethylamine, TEA; II, sodium bicarbonate, NaHCO$_3$; III, 1,8-diazabicyclo [5.4.0] undec-7-ene, DBU; IV, dimethylphenylphosphine, PMP; V, hexylamine, HA) as an initiator in polymerization. The normalized fluorescent spectra in Fig. 2c, d show that the continuous red shift in the emission $\lambda_{max}$ of PM powders from 465 nm (A-PM-HA) to 608 nm (A-PM-TEA) and solutions from 465 nm (A-PM-HA) to 624 nm (A-PM-TEA) occurs. Furthermore, the fluorescence intensities of PM solutions and photoluminescent quantum yield (PLQY) of PM powders decrease exponentially along with the redshift of emission $\lambda_{max}$ (Fig. 2b, Supplementary Fig. 7 and Supplementary Table 2). Notably, a side emission band lower than 450 nm accompanies the main emission band of red fluorescent PMs (e.g., A-PM-TEA in Fig. 2c, d above 600 nm, which is oriented toward two typical luminescent centers of PMs.

For living anionic polymerization, reaction time determines propagation degree until the complete consumption of monomer. The in situ fluorescence spectra of the reaction solutions initiated by Lewis bases (e.g., HA and TEA) at 80 °C were monitored in real time to investigate their emission changes with reaction time. Two distinct fluorescence variations over time were observed when HA and TEA were used. As shown in Fig. 2e, the fluorescence intensity of the HA-initiated system increases with prolonged reaction time and then rapidly deteriorates after reaching the peak intensity at 110 min. Accordingly, the fluorescent images inserted in Fig. 2e show the brightest blue emission in the midterm stage of the reaction. A slight redshift of the emission $\lambda_{max}$ occurs simultaneously. The unimodal feature of the emission spectra indicates that HA-initiated PM contains a single type of luminescent center or species. As presented in Fig. 2f, a similar rise and fall is observed in the emission intensity of TEA-initiated PM (A-PM-TEA) at ~450 nm. However, a blue

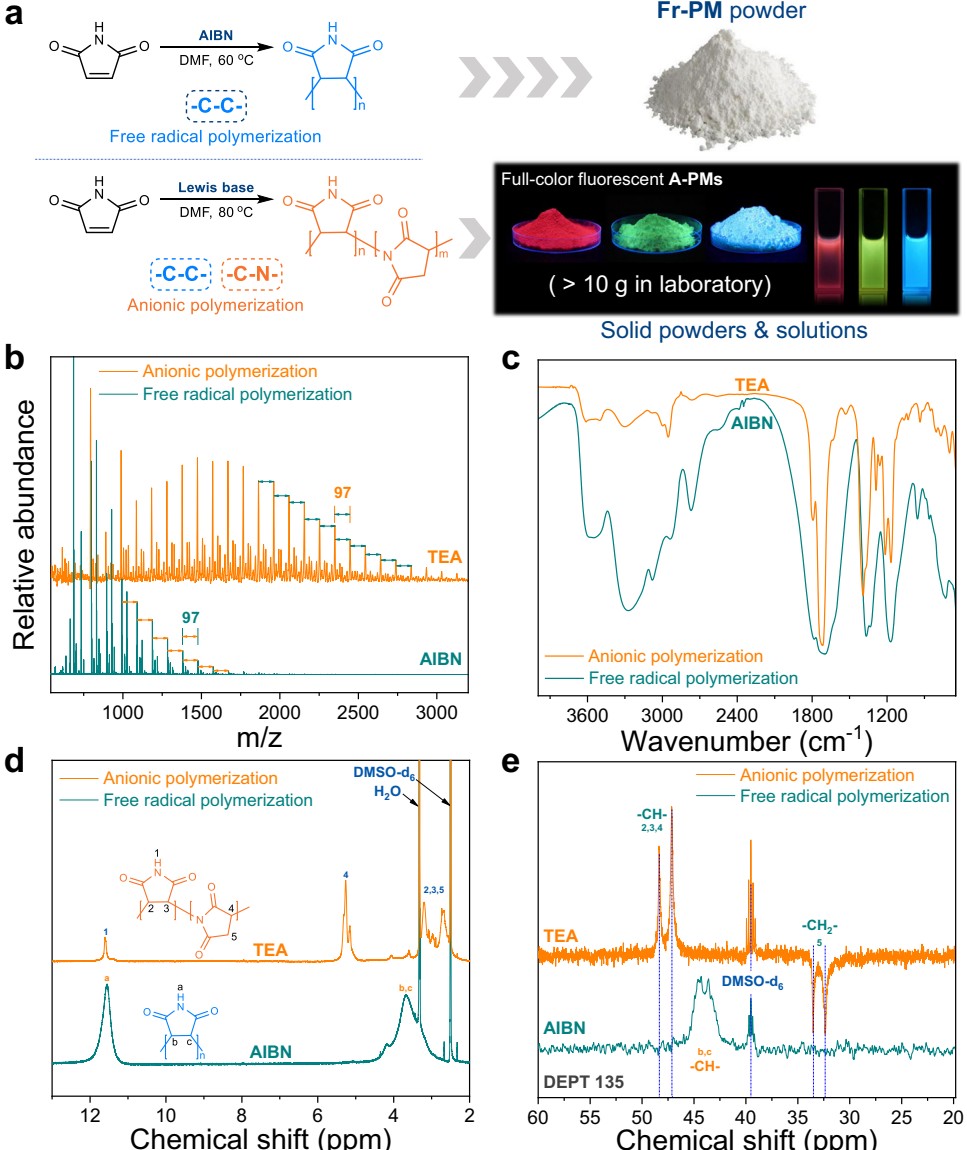

**Fig. 1 Synthesis and characterization of fluorescent polymaleimides (PMs). a** Synthetic routes and fluorescent images of PM powders and solutions in DMF via anionic polymerization (A-PM) and free-radical polymerization (Fr-PM). **b–e** MS, FTIR, $^1$H NMR and $^{13}$C NMR DEPT (Fr-PM was tested at 100 ºC) for comparative analysis of the typical A-PM and Fr-PM samples.

shift of the emission $\lambda_{max}$ (~450 nm) is observed, which is completely opposite to the results for HA-initiated PM. In addition, a new emission peak (~600 nm) appears, and a redshift of $\lambda_{max}$ along with a gradual increase in the $I_{600}/I_{450}$ (emission intensity) ratio occurs when reaction time is prolonged. Correspondingly, the fluorescent images inserted in Fig. 2f exhibit an obvious color transition from blue to red; i.e., the emission of PM can be regulated by controlling reaction time. In addition, the binodal emission spectra also suggest that two different types of chromophores or emission centers coexist in the TEA-initiated PMs. It can be further confirmed by the GPC results of A-PM-TEA (Supplementary Fig. 8), in which there are two distinct peaks of PMs corelated with different molecular weights. Because reaction time determines the molecular weight of PMs, a similar varying tendency of emission, including the emission intensity, wavelength ($\lambda_{max}$) and PLQY (%) is obtained. i.e., with the growth of PMs molecular weight (Supplementary Table 3 and Fig. 9), the emission intensity and PLQY rise first and decline afterward. For PMs synthesized with different Lewis bases, molecular weight is

also a major factor in their nonconventional luminescence (Supplementary Fig. 4 and Supplementary Tables 1 and 2). A larger molecular weight of PMs from anionic polymerization (initiated with Lewis bases) generally correlates with a longer emission wavelength ($\lambda_{max}$) and a lower PLQY. Unexpectedly, the molecular weight of PM from free-radical polymerization (Fr-PM-AIBN) is much larger than that of all PMs (e.g., A-PM-TEA) from anionic polymerization (Supplementary Table 1). However, Fr-PM-AIBN only has blue emission ($\lambda_{max}$). Such solid evidence once again proves that polymer structures, especially the linkage modes of repeat units (e.g., –C–C–, –C–N–), are as crucial as a molecular weight to the emission of PMs.

Reaction temperature is another variable that affects polymerization. Due to the low activation energy of initiation[49], anionic polymerization of maleimides even can be conducted at room temperature. In addition, the emission of PMs is not susceptible to the reaction temperature due to the fast initiation of anionic polymerization. As seen in Supplementary Fig. 10, there is no significant difference between the TEA-initiated PMs

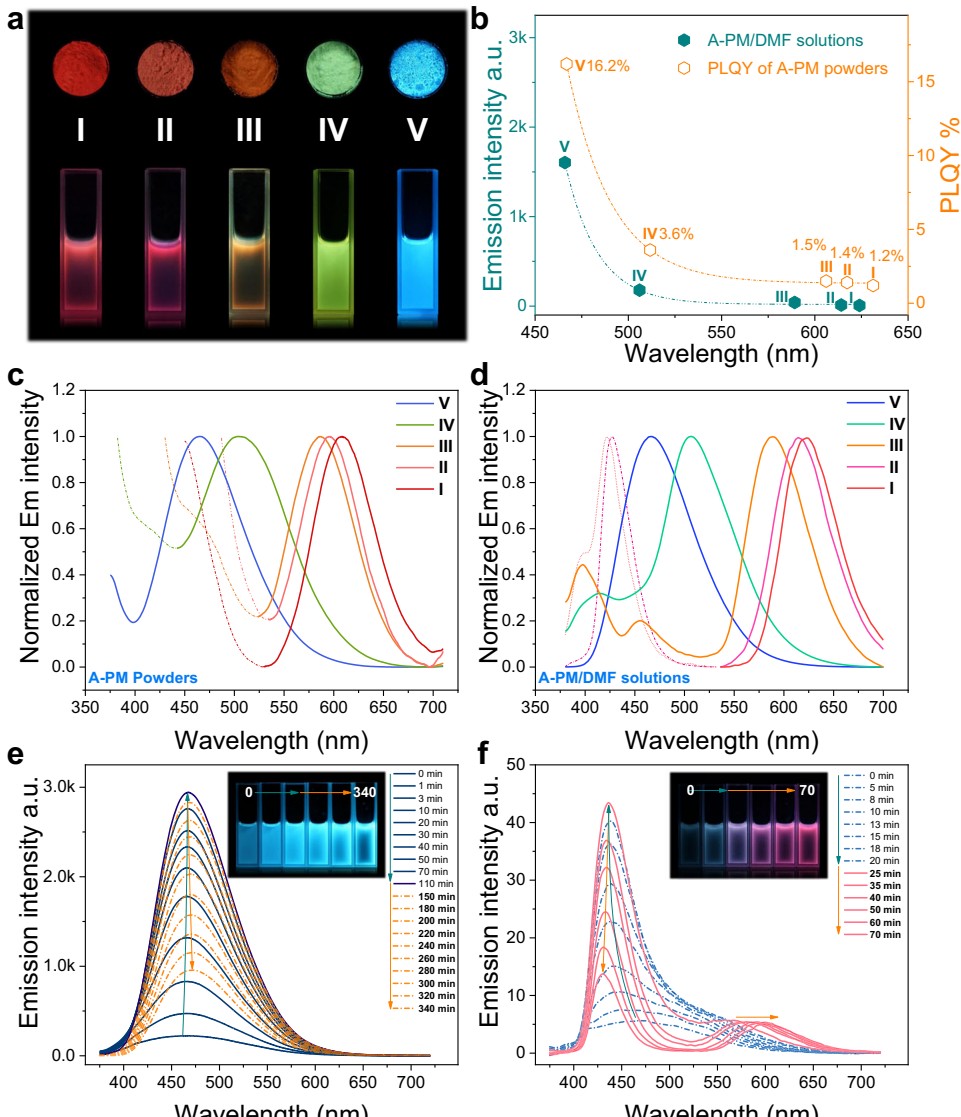

**Fig. 2 Full-color emission of A-PM regulated by anionic polymerization. a** Fluorescent images of A-PM powders and solutions in DMF synthesized with different Lewis bases (I, triethylamine, TEA; II, sodium bicarbonate, NaHCO₃; III, 1,8-diazabicyclo [5.4.0] undec-7-ene, DBU; IV, dimethylphenylphosphine, PMP; V, hexylamine, HA). **b** PLQY (%) of A-PM powders and emission intensity of A-PM/DMF solutions at $\lambda_{max}$. **c, d** Normalized emission spectra of A-PMs powders and solutions. The concentrations of I, II, III, IV, and V in solutions were 3.5, 7.0, 1.1, 25.0, and 25.0 mg/mL, respectively. **e, f** Emission spectra variations of maleimide solutions initiated by HA and TEA respectively with the prolongation of reaction time at 80 °C.

prepared at room temperature and those prepared at 80 °C. Such mild polymerization conditions make it possible for the large-scale production (Supplementary Fig. 2) and application of fluorescent PMs in practice. In summary, the emission color (or $\lambda_{max}$) of PMs can be finely tuned by altering their polymeric structures, which is controlled by different initiating species and reaction times. Eventually, full-color emission of nonconjugated PMs is realized.

**Concentration-dependent emission of A-PM.** CTE is presently the most prevalent nonconventional luminescent system. The general characteristic of concentration-enhanced emission, is a fundamental criterion of CTE. Therefore, the concentration-dependent emission of TEA-initiated PM is thoroughly investigated. As shown in Fig. 3a, b, when gradually increasing the concentration of A-PM-TEA/DMF solutions, the emission intensity at ~450 nm increases first and decreases after the

concentration reaches 3.56 mg/mL, accompanied with a blueshift of the emission $\lambda_{max}$ (lower than 450 nm). Notably, a new emission band (the enlarged view in Fig. 3a) appears at ~600 nm with a similar rise and fall of emission intensity (the inset in Fig. 3b). Opposite to the blue shift of $\lambda_{max}$ in the short-wavelength direction (~450 nm), a red shift of $\lambda_{max}$ at ~600 nm is observed. Interestingly, such concentration-dependent emission of A-PM-TEA is almost identical to the emission variation of A-PM-TEA with prolonging reaction time in Fig. 2f. It indicates that prolonging reaction time equivalently increases the concentration of PM in solutions. The concentration-dependent emission of A-PM-TEA is completely contradictory to the general characteristics of CTE and AIE luminescence. Instead, it is more similar to a luminescent characteristic of ACQ luminogens due to the inner-filter effect (Supplementary Fig. 12). Therefore, more theoretical assumptions and experimental verifications are needed to reach a universal mechanism of nonconventional luminescence, especially for CTE systems.

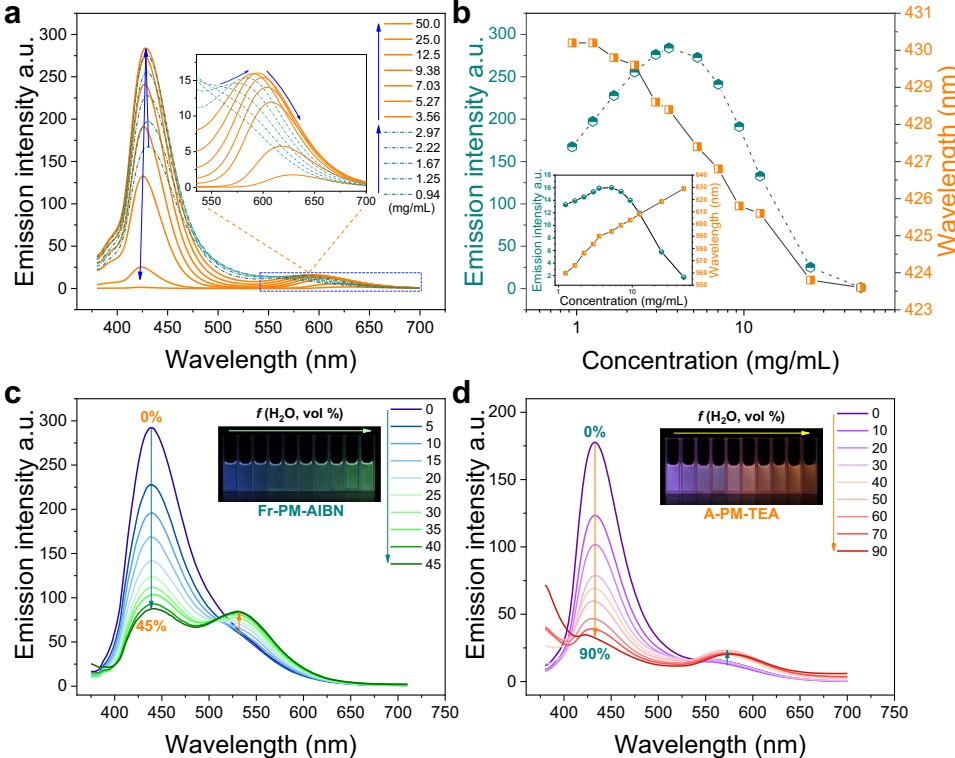

**Fig. 3 Concentration-dependent and solvent-dependent fluorescent properties of PM. a** Concentration-dependent emission spectra (inset plots are the enlarged view of spectra, 535–700 nm) of A-PM-TEA/DMF solutions. **b** maximum intensity and $\lambda_{max}$ changes around 425 nm (inset plots are the maximum intensity and $\lambda_{max}$ changes around 600 nm) of A-PM-TEA/DMF solutions with different concentrations. **c, d** Solvent-dependent emission spectra of Fr-PM-AIBN (100 mg/mL) and A-PM-TEA (5 mg/mL) solutions (DMF, good solvent) with different water contents (bad solvent).

**Solvent-dependent emission of A-PM.** To further confirm the luminescence characteristics of PMs, a typical experiment for AIE luminogens is implemented to investigate the emission of PMs in different aggregation states. As shown in Fig. 3c, d, $H_2O$, a poor solvent for Fr-PM-AIBN and A-PM-TEA, is added in PM/DMF solutions. A decrease in the emission intensity of both Fr-PM-AIBN and A-PM-TEA solutions at ~450 nm occurs with a slight blueshift of $\lambda_{max}$ upon gradually increasing water content (Fr-PM-AIBN is completely precipitated when water content exceeds 45 v/v%). This phenomenon is completely against the characteristics of CTE or AIE, more like ACQ. Furthermore, a shoulder emission band appears at a longer wavelength (Fr-PM-AIBN, 525 nm; A-PM-TEA, 575 nm). As a result, the emission color of Fr-PM-AIBN and A-PM-TEA solutions (inset images in Fig. 3c, d) gradually turns into green and red, respectively. Moreover, the related emission intensity of the shoulder peaks increases with the continuous addition of water, which coincides with the characteristics of CTE or AIE. Similar phenomena are observed when adding methanol to the A-PM-TEA/DMF solution (Supplementary Fig. 13). Herein, a paradox appears in the mechanism of nonconventional luminescent PM. Given the ACQ-like characteristics of PMs shown in Fig. 3a, b and a slight decrease in the emission intensity at 575 nm when water content is over 90 v/v% in Fig. 3d, it can be concluded that the luminescence of non-conjugated PMs should not be mechanically ascribed to CTE theory as usual. i.e., concentration-enhanced luminescence should no longer be regarded as a general characteristic of CTE. Furthermore, the high similarity of the bimodal spectra in Fig. 3a, c, d demonstrates that increasing concentration or adding a poor solvent to PM solutions changes the aggregation states associated with two types of luminescent centers or species with different emission wavelengths. As the aggregates of PMs gradually change,

emission color transitions to a longer wavelength are observed in Fig. 3c, d. In addition, the enhanced polarity obtained by adding water or methanol also contributes to the solvatochromism of PMs in solutions. The redshift in emission and decrease in intensity in such scenarios always point to the intramolecular (through-bond) or intermolecular (through-space) charge transfer (TBCT, TSCT)[50,51].

**Pressure-dependent emission and luminescent mechanisms of A-PM.** For a deep understanding of the nonconventional luminescence of PMs, the pressure-dependent emission of a typical PM from anionic polymerization (A-PM-TEA) is investigated with a diamond anvil cell (DAC). As shown in Fig. 4a, the emission of A-PM-TEA is gradually quenched when gradually increasing hydrostatic pressure from 0.7 to 11.7 GPa. Accordingly, an apparent red shift of $\lambda_{max}$ and a sharp decline of intensity are observed (Fig. 4b, d). After slowly withdrawing the pressure from 11.7 to 0.7 GPa, the emission $\lambda_{max}$ (Fig. 4c) and intensity (Fig. 4d) are completely restored to their initial states. In addition, as seen in Fig. 4g, the ratio of intensities at 627 and 595 nm, $I_{627}/I_{595}$, rises and falls linearly during the increase and decrease of pressure respectively. Such a linear dependence of $I_{627}/I_{595}$ and pressure suggests that the reversible variation of emission intensity is caused by molecular deformation and recovery under high pressure.

Given that the spectral shape has changed along with the apparent red shift of $\lambda_{max}$ under high pressure, multi-peak fitting analysis (Supplementary Figs. 16, 17) of the emission spectra has been conducted to reveal the luminescence mechanism of PMs under high pressure. Typical multi-peak fitting results in Fig. 4h suggest that there are at least three peaks (~580, 620, 650 nm)

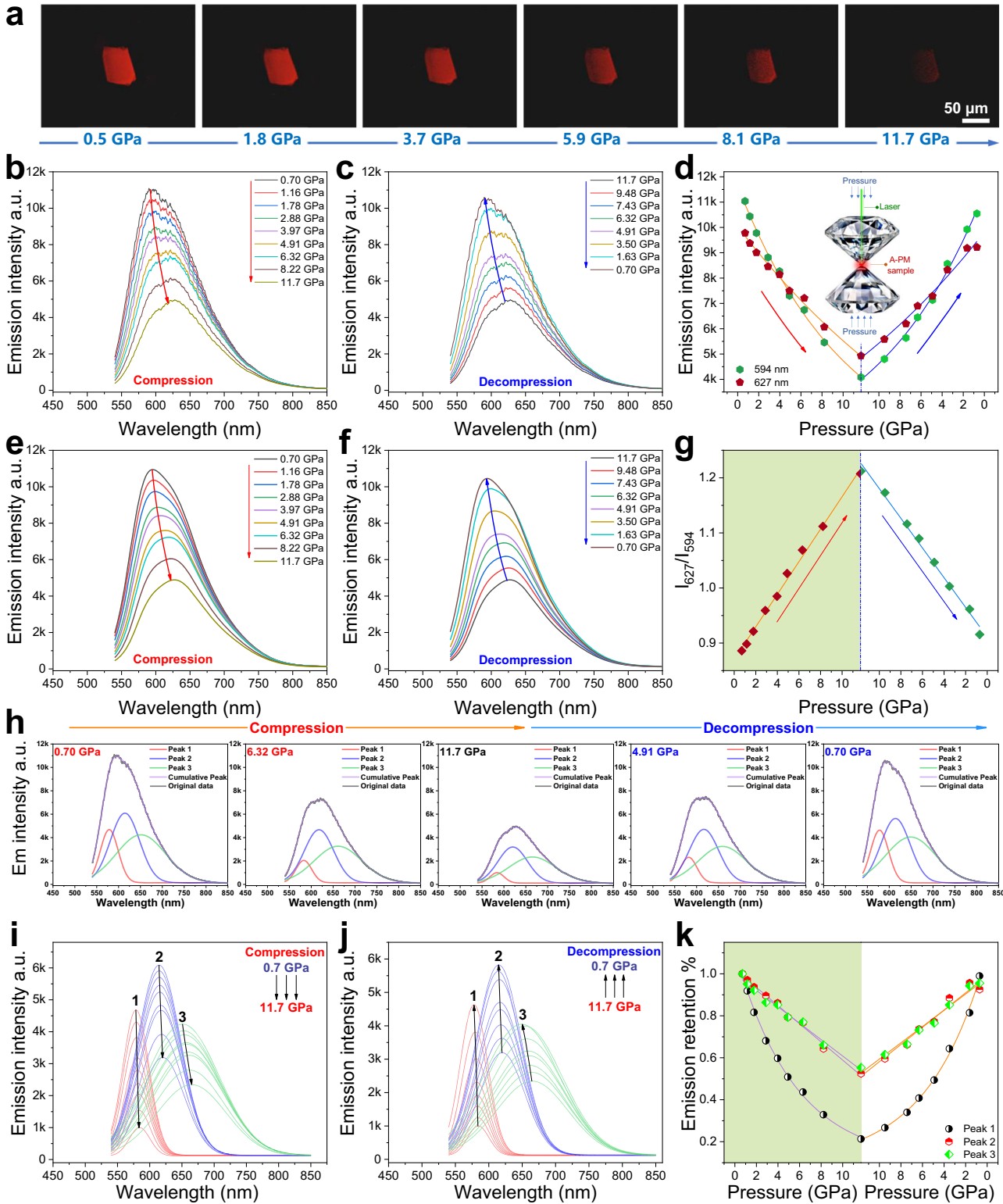

**Fig. 4 Pressure-dependent fluorescent properties of A-PM. a** In situ fluorescent images of A-PM-TEA compressed by different pressures. **b**, **c**, **e**, **f** Emission spectra and the multi-peak fitting plots of A-PM-TEA during compression (0.70–11.70 GPa) and decompression (11.70–0.70 GPa). **d**, **g** Emission intensities at different wavelengths (594 nm, $I_{594}$; 627 nm, $I_{627}$) and their ratio $I_{627}/I_{594}$ under different hydrostatic pressures. **h–k** Multi-peak fitting analysis, the peaks separated from the emission spectra of A-PM-TEA and the intensity variation of the peaks during compression and decompression.

beneath the emission spectra of A-PM-TEA under high pressure. Three separated emission peaks (1, 2, 3) are gradually quenched during compression (0.7–11.7 GPa, Fig. 4i) and fully recovered after decompression (11.7–0.7 GPa, Fig. 4j). Meanwhile, a recognizable red shift of $\lambda_{max}$ with rising the pressure (Supplementary Fig. 18) and a blue shift of $\lambda_{max}$ when releasing the pressure (Supplementary Fig. 19) are observed for three peaks. However, the largest shift $\lambda_{max}$ (<16 nm) of peak 3 is much smaller than the apparent $\lambda_{max}$ shift (>30 nm) of the spectra in Fig. 4b, c. Obviously, such apparent $\lambda_{max}$ shift should not be caused by the $\lambda_{max}$ shift of three peaks. As shown in Fig. 4i–k, comparatively, peak 1 is much more sensitive to pressure and it changes faster (larger slope in Fig. 4k) than peak 2 and 3 during the compression and decompression processes. Therefore, the apparent $\lambda_{max}$ shift in Fig. 4b, c is an accumulative result (Fig. 4e, f) of competition between the peaks with different varying rates of emission intensity under pressure. The characteristic peaks (580, 620, 650 nm) are associated with at least three different A-PM-TEA aggregates (e.g., different packing densities) under high pressure. The molecular deformation during compression reduces intermolecular distances. Generally, a closer intermolecular distance under high pressure is beneficial to through-space charge transfer (TSCT). Enhanced TSCT and ICT can significantly reduce the HOMO/LUMO energy gap ($\Delta E$). According to the equation between the rate of internal conversion ($k_{ic}$) and the energy gap $\Delta E$[52],

$$k_{ic} = 10^{13} e^{-a\Delta E} \tag{1}$$

A smaller $\Delta E$ gives a higher $k_{ic}$ and can intensify the non-radiative decay of the excited singlet state because of the coupling of rotational and vibrational energy levels. As a result, a red shift of $\lambda_{max}$ and a decline of intensity are observed for A-PM-TEA under high pressure.

In summary, according to the unique luminescence of PMs with distinct polymer structures (–C–C– and –C–N–) under different circumstances, a clear mechanism can be established to explain the nonconventional luminescence of PMs. The non-conventional full-color emission of PMs originates from different degrees of the intramolecular through-bond and/or intermolecular through-space charge transfer, TBCT and TSCT, which are mainly determined by the linkage mode (–C–C– and –C–N–), molecular weight and aggregation state of PMs. As illustrated in Fig. 5a, –C–C– and –C–N– are two typical linkage modes of repeat units in PMs. For the PMs mainly in –C–C– connection (e.g., Fr-PM-AIBN), the optimized electronic structures in Fig. 5b show that the electron cloud is mainly distributed on the periphery imide (O=C–NH–C=O) groups with strong electron-withdrawing ability. Through-bond charge transfer can occur in every isolated D–A unit (succinimide) but it is difficult to take place along the electron-deficient –C–C– backbone. As a result, –C–C– connected PM presents a relatively lower polarity (1.1182 Debye, 6 units) and an electron-rich periphery which is unfavorable for the close packing of PMs and intermolecular TSCT due to electrostatic repulsion. In comparison, for the PMs mainly in –C–N– connection (e.g., A-PM-TEA), electron cloud is alternatingly distributed on the imide (O=C–N–C=O) groups along the –C–N– backbone and a continuous D–A–D–A sequence is formed between the alternating electron-rich imide and electron-deficient tertiary –CH–. Charge transfer can be achieved through the –C–N– backbone with alternating D–A–D–A sequence, which is like a cascade pump and can significantly extend electron delocalization. Therefore, –C–N– connected PM has a comparatively higher polarity (9.7558 Debye, 6 units) and an asymmetric –C–N– backbone with alternating D–A–D–A units that is favorable for a staggered close packing of PMs and the intermolecular TSCT in aggregation similar to

head-J-aggregates. The result of the theoretical calculation (Supplementary Fig. 21) suggests that –C–N– connected PM (e.g., A-PM-TEA) with boosted intramolecular TBCT and intermolecular TSCT possesses a smaller HOMO/LUMO gap ($\Delta E$) than –C–C– connected PM (e.g., Fr-PM-AIBN). Therefore, A-PM-TEA shows a longer $\lambda_{max}$ of emission and a lower PLQY than Fr-PM-AIBN, which is in agreement with $\lambda_{max}$ red shift and intensity decline of PMs under high pressure (Fig. 4). Furthermore, according to the equation of PLQY ($\Phi$)[52] as follows,

$$\Phi = k_f/(k_f + k_{ic}) \propto 1/k_{ic} \tag{2}$$

$$\Delta E = h\nu = h/\lambda \tag{3}$$

$$\Phi \propto 10^{13} e^{ah/\lambda} \tag{4}$$

There is an exponential decay relationship between PLQY ($\Phi$) and emission $\lambda_{max}$ of PMs, which exactly coincides with the exponential decay curve of PLQY for PMs with different $\lambda_{max}$ in Fig. 2b.

Molecular weight effect on emission, abnormal concentration-, and solvent-dependent emission (compared with CTE) of PMs also can be rationally explained using the mechanism illustrated above. For example, the molecular weight effect on the emission of A-PM-TEA (Fig. 2f), at first stage of reaction, limited intramolecular TBCT is responsible for the blue emission band (450 nm) because only short D–A–D–A sequence (small molecular weight) is formed and A-PM-TEA has not reached its critical concentration for aggregation yet. Therefore, the blue emission of A-PM-TEA rises with the increase of molecular weight. With prolonging reaction time or proliferation of molecular weight, intramolecular TBCT and intermolecular TSCT are significantly enhanced due to the extended D–A–D–A sequence (large molecular weight) and formation of aggregates. Hence, a new red emission band (600 nm) appears. The emission intensity gradually declines with the growth of molecular weight, accompanying a continuous red shift of $\lambda_{max}$. In contrast, red shift for A-PM-HA over reaction time (Fig. 2e) is not very obvious because the restricted TBCT/TSCT in the isolated D–A units and the loose aggregates. In addition, if the succinimide units of A-PM-TEA is hydrolyzed in NaOH (aq.) (Supplementary Fig. 22) and the D–A–D–A sequence for intermolecular TBCT is broken, a blue emission (450 nm) like A-PM-HA is generated. It confirms that enhanced TBCT along –C–N– connected backbone is an essential attribute for the red emission of A-PM-TEA. Moreover, no obvious excitation-dependent luminescence of PMs is observed because of their relatively explicit luminescent species (Supplementary Fig. 23).

**Applications.** Excellent biocompatibility is the greatest advantage of nonconventional luminescent materials over aromatic luminogens. As shown in Fig. 6a, PMs show good skin-contact compatibility according to the HaCaT cell viability test. Therefore, PMs can be directly used in the scenarios requiring contact with human bodies in safety, e.g., security printing on banknote. As demonstrated in Fig. 6b, Security printing with specially designed patterns under natural light and UV 365 can be easily performed using screen-printing technology in lab with inks from trichromic PMs. The imide groups (O=C–NH–C=O) with abundant lone-pair electrons in PMs have a great coordination ability with metal ions that possess empty $p$ orbitals. As shown in Fig. 6c, the emission of A-PM-TEA/DMF solution can be selectively quenched by $Fe^{3+}$ due to the synergy between the inner-filter effect and static quenching mechanism (Supplementary Fig. 25). Thus, visualized recognition and detection of metal ions ($Fe^{3+}$) can be achieved on the basis of the color change and

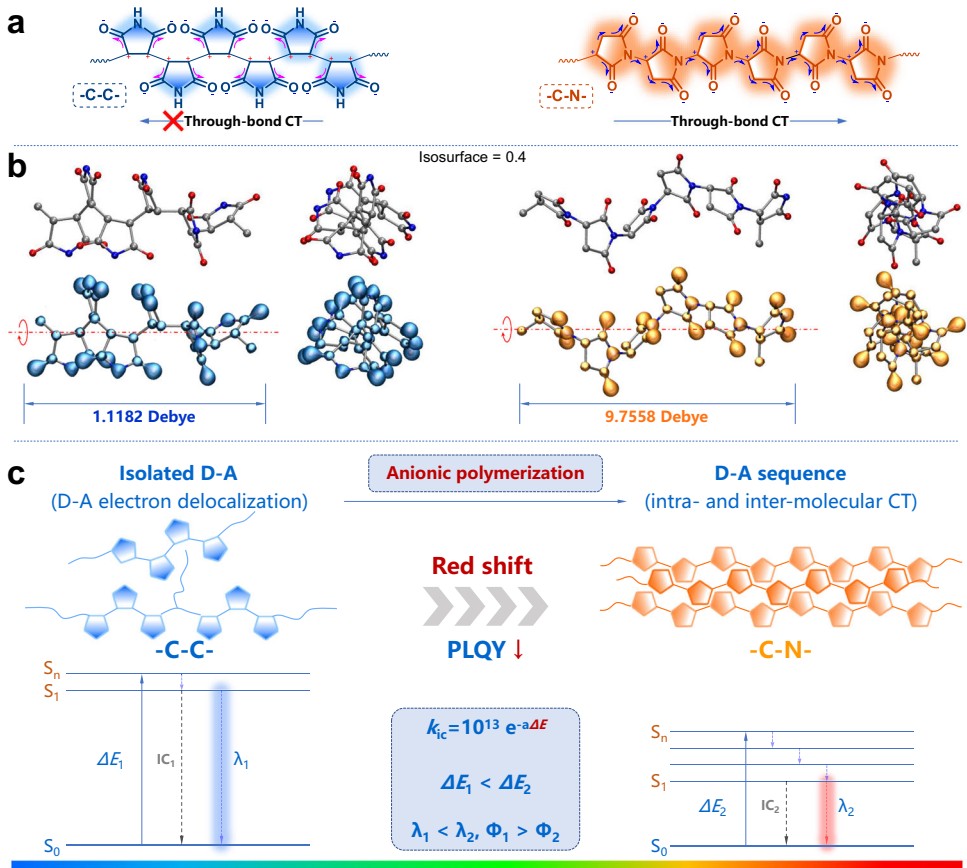

**Fig. 5 Schematic illustration of the nonconventional luminescence of PMs. a, b** Typical molecular structures, the optimized electronic structures and electron density distribution of the PMs in –C–C– and –C–N– connections calculated using Gaussian 09 with uB3LYP/6-31 G (**d**) method. **c** Schematic illustration the redshift of emission spectra and PLQY drop from –C–C– connected PM and –C–N– connected PM.

emission intensity variation to $Fe^{3+}$ with different concentrations. In particular, forensic fingerprint identification[53] on different substrates is accomplished with ultrafine PM powders because of the strong interactions between highly polar imides and fingerprint residues (e.g., proteins, amino acids). As shown in Fig. 6d, level 1 (patterns and ridge flows, e.g., whorls, arches, right and left loops), level 2 (ridge ending, bifurcation), and even level 3 (including scars and sweat pores, 1–9 in the fluorescence image) details of fingerprints can be identified in the high-solution fluorescence images, which is expected to greatly improve the accuracy of fingerprint identification in practice.

## Discussion

In summary, a new species of nonconventional luminescent material without any aromatics, poly(maleimide)s, is synthesized by the anionic polymerization of maleimide. Full-color emission is achieved by continuously tuning the emission spectra of PMs under different polymerization conditions (even at room temperature). However, typical CTE characteristics, such as concentration-enhanced emission and excitation-dependent luminescence, are not observed in PMs. In comparison with the limited through-space conjugation in the heteroatom clusters of the typical nonconjugated luminogens that have been reported, PMs have extended electron delocalization through intra- and/or intermolecular charge transfer, which can be regulated by the linkage mode (–C–C– and –C–N–), molecular weight and aggregation state of PMs in polymerization under different conditions (e.g., initiator, reaction time). Such biocompatible luminogens synthesized from a low-cost monomer contribute a unique example to help perfect CTE theory and provide a new

platform for rational molecular design to achieve full-color nonconventional luminescence far beyond their potential applications in security printing, fingerprint identification, metal ion recognition, etc.

## Methods

**Materials and reagents**. Maleimide, triethylamine (TEA), dimethylphenylphosphine (PMP), 1,8-diazabicyclo [5.4.0] undec-7-ene (DBU), hexylamine (HA), sodium bicarbonate ($NaHCO_3$), all solvents, metal ($Fe^{3+}$, $Mg^{2+}$, $Ca^{2+}$, $Ba^{2+}$, $Cr^{3+}$, $Mn^{2+}$, $Cu^{2+}$, $Cd^{2+}$, $Co^{2+}$, $Ni^{2+}$, $K^+$) chlorides, metal ($Fe^{2+}$) sulfate, metal ($Ag^+$) nitrate are commercially available. All chemicals were of analytical grade and were used without any purification. 2,2'-azobis(2-methylpropionitrile) (AIBN) was recrystallized with ethanol before use.

**Characterization instruments and methods**. [1]H NMR and [13]C NMR DEPT spectra were measured by an AVANCE 400 M (Bruker) NMR spectrometer in DMSO-$d_6$ or $D_2O$ (internal reference tetramethyl silane) at room temperature (sample conc. 10 mg/mL for [1]H NMR and 60 mg/mL for [13]C NMR). FTIR spectra were recorded by a Nicolet 6700 (Thermal Fisher) spectrometer using KBr-disk method (sample loading, 2 wt.%). MS spectra were obtained through an Autoflex III MALDI-TOF-MS (Bruker) time-of-flight mass spectrometer. Fluorescence spectra were acquired with a F-7000 fluorescence spectrometer (HITACHI) and UV-Vis spectra were measured with a Shimadzu UV 2600 system (ISR-2600Plus with an integrated sphere).

**Synthesis of A-PMs**. Maleimide (4.85 g) was dissolved in 50 mL of DMF, then one of the following Lewis bases, TEA, PMP, DBU, $NaHCO_3$, HA, was added into the solution as initiator of anionic polymerization. In detail, the feeding ration was 1:10 (molar ratio of initiator to monomer) when TEA, PMP, DBU or $NaHCO_3$ were used, while the ratio was 1:1 for HA. Subsequently, the solution was heated to 80 °C and stirred for different time (Supplementary Table 5). Purification procedures were quite different according to the initiator agent used. For TEA, PMP, and DBU, solid powders were obtained by directly precipitating the reaction solution in ethanol (washed with ethanol for three times and dried in a vacuum at 80 °C).

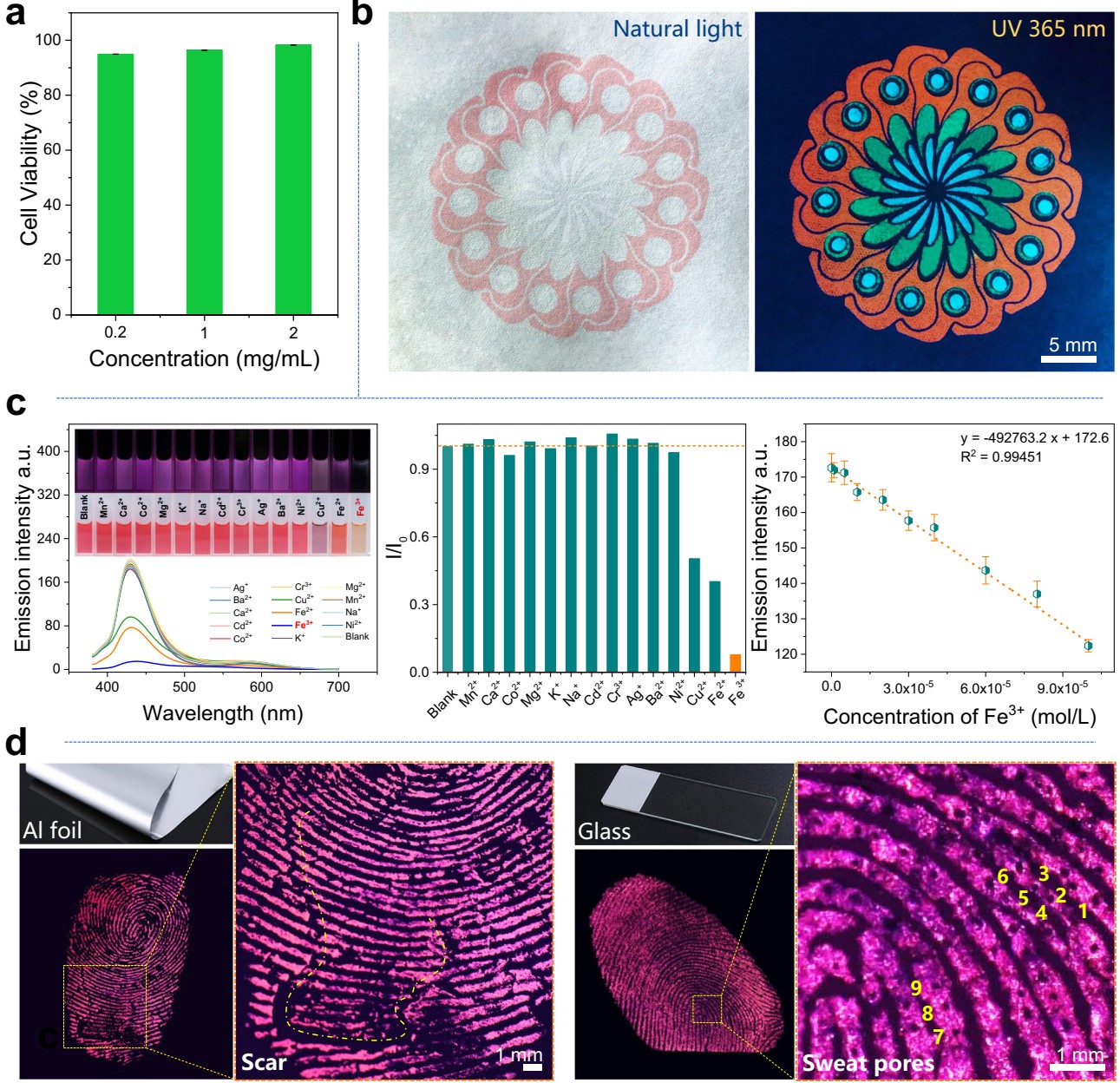

**Fig. 6 Biocompatibility and applications of A-PM. a** CCK-8 cell viability in the extracted liquid from A-PM-TEA powder. **b** Security printing by trichromic PMs inks under the natural light and UV 365 nm, respectively. **c** Fluorescent responses of A-PM-TEA/DMF to different metal ions and detection of $Fe^{3+}$ (excited at 365 nm), where the error bars indicate the standard deviation of the replicated tests (×3). **d** Forensic fingerprint identification on different substrates with an ultrafine A-PM-TEA powder.

For $NaHCO_3$, the reaction solution was dialyzed (dialysis tube, 100 Da) to remove the unreacted monomers and initiator first and then the solid sample was obtained by freeze-drying. For the reaction solution with HA, solid powder was recovered by directly precipitating the reaction solution in water (washed with ethanol for three times and dried in a vacuum at 80 °C), The product yield for different A-PMs is listed in Supplementary Table 5.

**Synthesis of Fr-PM.** Maleimide (6.07 g) was dissolved in 10 mL of DMF and AIBN (0.02 g) was added as initiator of free-radical polymerization. The solution was stirred in a nitrogen atmosphere for 30 min to remove oxygen and then heated to 60 °C and stirred for 24 h. Finally, the solution was precipitated in ethanol. The filtered precipitate was washed with ethanol for three times and then dried in vacuum at 80 °C to obtain white powders (yield = 72 %).

**Pressure-dependent fluorescence of A-PM.** In total, 2.0 g of A-PM-TEA powder was dissolved in 5.0 mL of DMF. The solvent was slowly evaporated on a hot plate

(80 °C) in order to eliminate the voids in A-PM-TEA powders and a solid sample like crystal was finally obtained.

The pressure-dependent emission of A-PM-TEA was investigated using a diamond anvil cell (DAC, IIa). The culet diameter of a diamond is 500 μm. A T301 gasket (original thickness 250 μm) was preindented to 50 μm and a central hole (diameter, 200 μm) was drilled to load the solid sample of A-PM-TEA. Ruby spheres were inserted into the central hole for in situ pressure calibration and silicone oil was used to transmit the pressure. The photoluminescent spectra under high pressure were recorded with a confocal Raman microscope (HORIBA LabRAM HR Evolution) at room temperature. Instrument setting: ND filter = 0.1%, acq. time(s) = 0.1, accumulations = 60, grating = 600 (500 nm), excitation wavelength 532 nm and scan range 540–850 nm. The pressure was manually adjusted through a screw rod and balanced for 5 min. Then the fluorescent spectra and confocal images of A-PM-TEA under different hydrostatic pressures were acquired[54].

**Theoretical calculation study.** Herein, the optimized molecular structures and wavefunctions of PMs based on time-dependent density functional theory (TD-

DFT) were calculated using Gaussian 09[55] with uB3LYP/6-31 G(d) package. Multiwfn was utilized to obtain the electron densities, energy levels, and electronic potential distributions. VMD was used to visualize the HOMO and LUMO of PMs. The optimized molecular conformations and excited-state HOMO/LUMO of PMs are illustrated in Supplementary Figs. 20, 21, respectively. In comparison, –C–N– connected PM has a larger dipole moment and a smaller HOMO/LUMO energy gap than –C–C– connected PM, which exactly coincides with their luminescence characteristics.

## Data availability

All the data generated or processed in this study are presented in the article and Supplementary Information files.

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

## Acknowledgements

This work was supported by Beijing National Laboratory for Molecular Sciences (BNLMS-CXXM-202007) and the National Natural Science Foundation of China (Nos. 51803220 and 51773210.).

## Author contributions

Academic writing and conceptual design were finished by W.T. and X.J. Experiments and data analysis were performed by X.J. and K.J. H.D. and G.S. conducted the pressure-dependent fluorescence. X.H. assisted to undertake the theoretical study. This work was finished under the supervision of W.T. and J.Z.

## Competing interests

The authors declare no competing interests.
