## [Peer Review File · Nature Communications]

Anionic polymerization of nonaromatic maleimide to achieve full-color nonconventional luminescenceReviewers' Comments:

Reviewer #1:

Remarks to the Author:

In this work, Ji et al. reported the nontraditional fluorescence in polymaleimides (PMs) which are without any aromatic rings, the polymers were synthesized by two methods: anionic polymerization (A-PM) and radical polymerization (Fr-PM). Interestingly, full-color emission (blue to red) is achieved by tuning polymerization conditions (catalyst, reaction time, reaction temperature, et al). The author also demonstrated their potential applications in security printing, fingerprint identification, metal ion recognition, The reported phenomenon is quite interesting. However, just like the authors' statement in the main text, such phenomenon has been preliminarily reported by many groups, the author's contribution in this work is to systematically investigate such phenomenon. Unfortunately, no detailedly mechanistic studies have been carried out in this work and the mechanism assumption raised by authors is also not convinced which shows a lot of drawbacks. Whereas, the mechanistic part is the crucial factor to determine whether this work is suitable to publish in Nat. Commun. or not because the novelty of the phenomenon itself is not high. Based on these considerations, the current version of this MS is not suitable to be accepted. However, I am glad to consider another resubmitted version once the authors address the following several major revisions.

1. Although a lot of PMs were reported in this work, the authors did not establish a clear structure-property relationship between the chemical structures and luminescent properties, especially from the quantitative perspective. They only mentioned or proposed that the blue and red emissions originate from the C-C and C-N linkages, respectively. So, the authors are suggested to characterize each of the polymers by NMR, MS, DSC, PL et al, then summarize all these data and quantitatively establish a clear structure-property relationship between polymer structure and nonconventional luminescence.
2. The authors proposed that the various emission colors are caused by the structural difference of these polymers which are produced by using different Lewis bases. However, as mentioned by previously reported works, the red emission is caused by the complex between maleimide units and the trace amount of Lewis bases (amine group), which has been proved by many works. So, how the authors could exclude the interference of such complex in the current systems as these catalysts are difficult to be completely removed from the polymers.
3. In figure 4d, how did the authors obtain the conformations of these two polymers? As mentioned above, the clear mechanism picture plays a decisive role in this work, but the authors almost haven't touched this part besides some preliminary speculation. So the authors are highly recommended to optimize the structures of these polymers via quantum mechanical calculation or molecular dynamic simulation, meanwhile, their excited-state electronic structures should also be calculated for the establishment of mechanistic picture.
4. In the introduction part, the authors are suggested to introduce the research frontier of mechanism for the nonconventional luminescence, especially for the PMs systems. Meanwhile, according to most of the reported works on CTE, CTE is the phenomenon of such effect but not the mechanism.
5. In figure 3a-b, the blue emission was quenched at high concentration, the authors are suggested considering the possibility of inner-filter effect.
6. In figure 2b, the authors have compared the emission intensity of the polymer powders and drawn the plots. However, it is almost impossible to compare the powder samples as the number of molecules excited by the excitation beam is different. So, the authors are suggested redoing these experiments by comparing their emission quantum efficiency instead of intensity.
7. In order to disclose the emission mechanism of these polymers, the variable temperature PL measurement (rt to ~100 oC) is suggested to carry out for the relevant PMs.
8. Line 281-283, the authors stated that "This suggests that deformation under hydrostatic pressure (< 5 GPa) reduces the intermolecular distance and that the enhanced molecular orbital overlap can promote intermolecular charge transfer." However, if the molecular orbital overlap is increased, why the emission wavelength keeps almost no change? Meanwhile, why the enhanced TSCT will activate the internal conversion?

Reviewer #2:

Remarks to the Author:

Nonconventional luminophores without aromatic rings have attracted much attention and become a hot topic of research nowadays. In this paper, the authors reported the nonconventional luminescent behaviors of nonaromatic poly(maleimide)s (PMs). They achieved full-color emission (blue, green, red) of PMs via anionic polymerization of nonaromatic maleimide catalyzed by different organic bases. The resultant PMs showed aggregation-caused quenching (ACQ) instead of cluster-triggered emission (CTE) characteristic which is usually observed in most nonconventional luminophores. Finally, some simple applications were demonstrated. The results are interesting. The manuscript is recommended for publication in Nature Communications after careful revisions.

(1) The nonconventional luminescent properties of maleimide derivatives from small molecules to polymers have been reported in recent years (e.g., Chem. Commun. 2015, 51, 9733; Macromolecules 2020, 53, 3756–3764; Adv. Sci. 2021, 8, 2004299). These relevant papers may be mentioned and cited in the Introduction section.

(2) Because the nonconventional luminescent behaviors of PMs reported by the authors are quite different from many reported works, some theoretical calculations are needed for deep understanding of these phenomena instead of only assumptions.

(3) The authors compared the emission intensity of A-PM powders, but it is hard to make all conditions the same for different solid powders during PL tests. Thus, the absolute photoluminescence quantum yields (PLQYs) of the polymers should be given for clear comparison.

(4) The molecular weights and their distributions of all polymers should be given.

(5) For the same catalytic system, the authors should study molecular weight effect on the emission color.

(6) The authors mentioned TSCT effect both in "Solvent-dependent emission" and "Pressure-dependent emission". The emission redshift and intensity decrease were both observed in "Solvent-dependent emission", but only intensity decrease was observed in "Pressure-dependent emission". Why did the emission wavelength keep the same with the increased pressure and TSCT effect?

(7) Please explain why enhanced TSCT would facilitate internal conversions.

(8) How did the authors confirm the concentration of the polymers in DMEM after filtered with a syringe-driven filter (0.2 μm) in "Cell viability test"? If only the original concentration was used, it would make no sense.

(9) The description of "Application" is too short. The Fe(III) quenching mechanism should be discussed.

(10) Some small mistakes: Figure 2, "(d) HA and (e) TEA" should be "(e) HA and (f) TEA"; Figure 5c, the colors of emission lines for different ions were marked wrongly.

Point-by-point Responses to Reviewers

Dear Dr. Johannes Kreutzer,

We would like to thank you for giving us adequate time to polish our work. We also would like to express our great gratitude to the kind comments of reviewers that really help improve our manuscript. As required, all the comments have been responded point-by-point. The revision has also been made and highlighted in the copies of manuscript and the supplementary information.

Reviewer #1 (Remarks to the Author)

In this work, Ji et al. reported the nontraditional fluorescence in polymaleimides (PMs) which are without any aromatic rings, the polymers were synthesized by two methods: anionic polymerization (A-PM) and radical polymerization (Fr-PM). Interestingly, full-color emission (blue to red) is achieved by tuning polymerization conditions (catalyst, reaction time, reaction temperature, et al). The author also demonstrated their potential applications in security printing, fingerprint identification, metal ion recognition, The reported phenomenon is quite interesting. However, just like the authors' statement in the main text, such phenomenon has been preliminarily reported by many groups, the author's contribution in this work is to systematically investigate such phenomenon. Unfortunately, no detailly mechanistic studies have been carried out in this work and the mechanism assumption raised by authors is also not convinced which shows a lot of drawbacks. Whereas, the mechanistic part is the crucial factor to determine whether this work is suitable to publish in Nat. Commun. or not because the novelty of the phenomenon itself is not high. Based on these considerations, the current version of this MS is not suitable to be accepted. However, I am glad to consider another resubmitted version once the authors address the following several major revisions.

1. Although a lot of PMs were reported in this work, the authors did not establish a clear structure-property relationship between the chemical structures and luminescent properties, especially from the quantitative perspective. They only mentioned or proposed that the blue and red emissions originate from the C-C and C-N linkages, respectively. So, the authors are suggested to characterize each of the polymers by NMR, MS, DSC, PL et al, then summarize all these data and quantitatively establish a clear structure-property relationship between polymer structure and nonconventional luminescence.

Figure S1. (a) ^1H NMR, (b) DEPT-135 ^{13}C NMR, (c) FTIR, and (d) MS spectra; (e) GPC, (f) DSC,

(g) TG, (h) DTG results of PMs synthesized with different initiators.

Table S1. Number-average molecular weight (Mn), weight-average molecular weight (Mw), polydispersity (Mw/Mn) of different PMs.

Sample	λ_{\max} , nm	Mn	Mw	Mw/Mn
A-PM-HA	464.8	2.053×10^2 ($\pm 8.719\%$)	2.078×10^2 ($\pm 8.492\%$)	1.01
A-PM-PMP	504.0	8.578×10^2 ($\pm 1.445\%$)	1.406×10^3 ($\pm 2.681\%$)	1.64
A-PM-DBU	586.6	9.286×10^3 ($\pm 1.024\%$)	1.062×10^4 ($\pm 1.171\%$)	1.14
A-PM-NaHCO ₃	596.4	8.844×10^3 ($\pm 1.446\%$)	1.452×10^4 ($\pm 1.257\%$)	1.64
A-PM-TEA	608.6	1.024×10^4 ($\pm 0.904\%$)	1.620×10^4 ($\pm 0.801\%$)	1.58
Fr-PM-AIBN	467.4	3.825×10^4 ($\pm 0.567\%$)	4.927×10^4 ($\pm 0.251\%$)	1.29

RESPONSE:

Sincere gratitude for your suggestion.

As suggested, we have systematically characterized each of PMs by ¹H NMR, DEPT-135 ¹³C NMR, FTIR, MS spectra, GPC, DSC, TG, DTG and try to establish a quantitative structure-property relationship between polymer structure and nonconventional luminescence. The results in Figure S1 and Table S1 indicate that polymerization mode and molecular weight are the major factors to determine the nonconventional luminescence of PMs. For example, a larger molecular weight of the PMs from anionic polymerization (initiated with Lewis bases) generally correlates with a longer emission wavelength (λ_{\max}). On the contrary, the molecular weight of PM (Fr-PM-AIBN) from free radical polymerization is much larger than that of each PM (e.g., A-PM-TEA) from anionic polymerization. However, Fr-PM-AIBN only has blue emission (λ_{\max}). Such solid evidence suggests that the polymerization mode (-C-C-, -C-N- or other repeat units) should be responsible for the difference between the emission wavelengths of Fr-PM-AIBN and A-PM-TEA.

Note: the related supplement and revision have been made and highlighted in the revised version of manuscript.

2. The authors proposed that the various emission colors are caused by the structural difference of these polymers which are produced by using different Lewis bases. However, as mentioned by previously reported works, the red emission is caused by the complex between maleimide units and the trace amount of Lewis bases (amine group), which has been proved by many works. So, how the authors could exclude the interference of such complex in the current systems as these catalysts are difficult to be completely removed from the polymers.

Figure S2. Fluorescent spectra of A-PM-TEA (solution and powder)/TEA mixtures at different TEA concentrations.

Table S2 PLQY of A-PM-TEA/TEA mixtures (powders) at different TEA concentrations.

TEA concentration, wt.%	PLQY %
50.0	0.5 %
10.0	0.4 %
1.0	0.5 %
0.1	0.4 %
0	0.4 %

RESPONSE:

Thanks a lot for your comment.

The potential complex between maleimide units and the trace amount of Lewis bases (amine groups) should be concerned to demonstrate red emission of the nonconventional luminescent PMs. However, the interference of such complex can be excluded from our systems on the basis of following experimental evidences.

1. The molecular structure and luminescent characteristics of A-PM-TEA and A-PM-NaHCO₃ (major repeat units, -C-N-; orange red emission) are very close. It is impossible that red emission of A-PM-NaHCO₃ originates from the maleimide-amine complex, because TEA is not involved in A-PM-NaHCO₃. In addition, different from the organic Lewis base TEA, the inorganic NaHCO₃ can be easily removed by water washing. Therefore, the interference of NaHCO₃ is also excluded.

2. If the red emission is caused by the complex between maleimide units and the trace amount of amines, the addition of TEA into A-PM-TEA solution or powders would enhance n red emission because of the increased maleimide-amine complexes. However, as seen in Figure S2, different amount of TEA was blended into A-PM-TEA/DMF solutions or A-PM-TEA powders and no obvious changes were observed in their emission spectra. The PLQY of A-PM-TEA/TEA mixture (powders) in Table S2 indicates that the addition of TEA has little influence on the emission of A-PM-TEA, considering the instrumental error for powder tests.

Above experimental results confirm that the red emission of A-PM-TEA should be attributed to its polymer structure rather than the residual maleimide-amine complexes.

3. In figure 4d, how did the authors obtain the conformations of these two polymers? As mentioned above, the clear mechanism picture plays a decisive role in this work, but the authors almost haven't touched this part besides some preliminary speculation. So, the authors are highly recommended to optimize the structures of these polymers via quantum mechanical calculation or molecular dynamic simulation, meanwhile, their excited-state electronic structures should also be calculated for the establishment of mechanistic picture.

Figure S3. Optimized molecular conformation and dipole moment (6 repeat units) of PMs in -C-C- and -C-N- connections calculated using uB3LYP/6-31G(d) method.

Figure S4. Excited-state HOMO/LUMO of PMs (6 repeat units) calculated using uB3LYP/6-31G(d) method.

RESPONSE:

Thanks very much for your advices.

In our manuscript, the conformations of -C-C- and -C-N-connected PMs were first optimized using the MM2 dynamics in Chem3D. As recommended, the optimized molecular structures and wavefunctions of PMs based on time-dependent density functional theory (TD-DFT) were calculated using Gaussian 09 with uB3LYP/6-31G(d) package. Multiwfn was utilized to obtain the electron densities, energy levels, and electronic potential distributions. VMD was used to visualize the HOMO and LUMO of PMs. The optimized molecular conformations and excited-state HOMO/LUMO of PMs are illustrated in Figure S3 and S4, respectively. In comparison, -C-N-connected PM has a larger dipole moment and a smaller HOMO/LUMO energy gap than -C-C-connected PM, which exactly coincides with their luminescence characteristics.

4. In the introduction part, the authors are suggested to introduce the research frontier of mechanism for the nonconventional luminescence, especially for the PMs systems. Meanwhile, according to most of the reported works on CTE, CTE is the phenomenon of such effect but not the mechanism.

RESPONSE:

Gratitude for your kind suggestions.

Mechanisms for the typical nonconventional luminescence has been added into the introduction section. Given there are no reports about the nonconventional luminescent polymers from the anionic polymerization of maleimide, the nonconventional luminescent polymers containing isolated maleimide moieties and poly(maleic anhydride)s have been summarized in the introduction section.

As suggested by reviewer, CTE should be a general item of the luminescent phenomenon triggered by clusterization instead of mechanisms. The related parts that may lead misunderstanding have been revised according to the reviewer's suggestion.

Note: the related supplement and revision have been made and highlighted in the revised version of manuscript.

5. In figure 3a-b, the blue emission was quenched at high concentration, the authors are suggested considering the possibility of inner-filter effect.

Figure S5. (a) UV-Vis, excitation and emission spectra of A-PM-TEA/DMF solution. (b) the fluorescence decay time plots of A-PM-TEA/DMF solutions at 1 mg/mL and 20 mg/mL.

RESPONSE:

Thanks for your kind suggestions.

As suggested, the excitation and emission spectra of A-PM-TEA/DMF solution at a higher concentration (20 mg/mL) were shown in Figure S5a. The partial overlaps between the excitation/absorption spectra (short-dot lines) and the emission spectra (solid lines) of A-PM-TEA confirm that secondary inner-filter effect is inevitable in A-PM-TEA/DMF solutions at a higher concentration. Additionally, there is no obvious change between the fluorescence lifetime plots (Figure S5b) of A-PM-TEA/DMF at 1 mg/mL and 20 mg/mL, which is a prerequisite characteristic of inner-filter effect. Therefore, the secondary inner-filter effect should be a major path responsible

for the emission quenching of A-PM-TEA at high concentration and the emission wavelength shift (red shift at long wavelength and blue shift at short wavelength) should result from the aggregation variation of A-PM-TEA at different concentrations.

By contrast, the emission changes in Figure 3c-d should not be caused by inner filter effect because the concentrations of all A-PM-TEA/DMF solutions are the same. The emission intensity decay at short wavelength and rise at long wavelength are simply from the aggregation change of luminophores caused by adding different amount of H₂O into the solutions.

6. In figure 2b, the authors have compared the emission intensity of the polymer powders and drawn the plots. However, it is almost impossible to compare the powder samples as the number of molecules excited by the excitation beam is different. So, the authors are suggested redoing these experiments by comparing their emission quantum efficiency instead of intensity.

Figure S6 Emission intensity and related PLQY of A-PM powders synthesized with different Lewis bases (I, triethylamine, TEA; II, sodium bicarbonate, NaHCO₃; III, 1,8-diazabicyclo [5.4.0] undec-7-ene, DBU; IV, dimethylphenylphosphine, PMP; V, hexylamine, HA).

Table S3 PLQY of A-PM powders synthesized with different Lewis bases ($\lambda_{\text{ex}} = 365 \text{ nm}$).

No.	Lewis base	PLQY %
I	TEA	1.2
II	NaHCO ₃	1.4
III	DBU	1.5
IV	PMP	3.6
V	HA	16.2

RESPONSE:

Thanks for your suggestions.

It is indeed very difficult to compare the luminescence of powders directly by their emission intensities because the uniform test conditions are almost impossible to achieve for powders. Thus, the absolute photoluminescence quantum yields (PLQYs) of the powders have been given in Figure S6 and Table S3 for clear comparison. Fortunately, the PLQYs variations coincide with the emission intensity plot of different A-PM powders.

7. In order to disclose the emission mechanism of these polymers, the variable temperature PL measurement (rt to ~100 oC) is suggested to carry out for the relevant PMs.

Figure S7. (a) Temperature-dependent emission spectra and (b) Normalized temperature-dependent emission spectra of A-PM-TEA/DMF solution (3 mg/mL, 25-95 °C).

RESPONSE:

Thanks for your suggestions.

The temperature-dependent emission of A-PM-TEA has been carried out from 25 to 95 °C. As seen in Figure S7a, a normal decrease of emission intensity with rising the temperature was observed along with a slight red shift of maximum emission wavelength, which is extremely similar to the results in pressure-dependent emission of A-PM-TEA powders.

The drop in emission intensity with rising temperature is generally ascribed to the intensified molecular motions (rotation, vibration, collision with solvent molecules etc.) that enhance nonradiative decays. Meanwhile, the slight red shift of emission wavelength should be attributed to enhanced solvent relaxation at a higher temperature that can lower the excited state of fluorophores and result in red shift in emission spectra.

8. Line 281-283, the authors stated that “This suggests that deformation under hydrostatic pressure (< 5 GPa) reduces the intermolecular distance and that the enhanced molecular orbital overlap can promote intermolecular charge transfer.” However, if the molecular orbital overlap is increased, why the emission wavelength keeps almost no change? Meanwhile, why the enhanced TSCT will activate the internal conversion?

Figure S8. Pressure-dependent emission of A-PM-TEA (0-5 GPa): (a, b) *in-situ* emission spectra of A-PM-TEA under different hydrostatic pressures. Normalized emission spectra and their enlarged views of A-PM-TEA when rising (c, d) and decreasing (e, f) the hydrostatic pressures.

Figure S9. Pressure-dependent emission of A-PM-TEA (0-10 GPa): Normalized emission spectra of A-PM-TEA when rising (a) and decreasing (b) the hydrostatic pressures.

RESPONSE:

Thanks for your comments.

First, with respect to the emission wavelength shift under hydrostatic pressure.

As noticed by reviewer, it is difficult to observe obvious changes of emission wavelengths (λ_{\max}) in Figure S8a-b (the original data in the first submitted manuscript). There are several reasons for the weak signals of emission wavelength shift.

- (1) The ratio of signal to noise (S/N) in the fluorescence spectra under high hydrostatic pressures is much lower than that in normal fluorescence spectra. In addition, emission wavelength (λ_{\max}) is generally not as sensitive as the emission intensity to pressure.
- (2) The decrease of intermolecular distance induced by a limited hydrostatic pressure (5 GPa) is insufficient to generate a significant red shift of emission λ_{\max} that can be directly observed.

In fact, a noticeable change of emission wavelengths under different hydrostatic pressures can be observed in the enlarged views of the normalized emission spectra Figure S8c-f. A slight red shift of the peaks at approximate 600, 635 and 660 nm appears and the shoulder peak at approximate 660 nm gradually rises when increasing the pressure to 5 GPa. Oppositely, a blue shift of the emission wavelengths and a fall of the shoulder peak at approximate 660 nm occur simultaneously. As shown in Figure S9, the above shift of λ_{\max} and variation of the shoulder emission around 660 nm become much more evident when increasing the maximum pressure to 10 GPa, in comparison with pressure-dependent emission of A-PM-TEA under hydrostatic pressures ranged from 0-5 GPa.

The positive results in Figure S9 encouraged us to repeat our experiment in Figure S8a-b under a large pressure range (0-12 GPa) with an advanced spectrometer. The experimental details are as follows.

2.0 g of A-PM-TEA powder was dissolved in 5.0 mL of DMF. The solvent was slowly evaporated on a hot-plate (80 °C) in order to eliminate the voids in A-PM-TEA powders and a solid sample like crystal was finally obtained.

The pressure-dependent emission of A-PM-TEA was investigated using a diamond anvil cell (DAC,

IIa). The culet diameter of diamond is 500 μm . A T301 gasket (original thickness 250 μm) was preindented to 50 μm and a central hole (diameter, 200 μm) was drilled to load the solid sample of A-PM-TEA. Ruby spheres were inserted into the central hole for in-situ pressure calibration and silicone oil was used to transmit the pressure. The photoluminescent spectra under high pressure were recorded with a confocal Raman microscope (HORIBA LabRAM HR Evolution) at room temperature. Instrument setting: ND filter = 0.1%, acq. time(s) = 0.1, accumulations = 60, grating = 600 (500 nm), excitation wavelength 532 nm and scan range 540-850 nm. The pressure was manually adjusted through a screw rod and balanced for 5 min. Then the fluorescent spectra and confocal images of A-PM-TEA under different hydrostatic pressures were acquired.

Figure S10. Pressure-dependent emission of A-PM-TEA (0-12 GPa): (a) *in-situ* fluorescent images and (b, c) emission spectra of A-PM-TEA under different hydrostatic pressures. (d, e) the emission intensities of A-PM-TEA at 594 nm and 627 nm and their ratio I_{627}/I_{594} under different hydrostatic pressures.

Multi-peak fitting analysis of the emission spectra during the pressure rise (0 to 12 GPa)

Figure S11. Multi-peak fitting results of the emission spectra of A-PM-TEA under the hydrostatic pressure increasing from 0 to 12 GPa.

Multi-peak fitting analysis of the emission spectra during the pressure fall (12 to 0 GPa)

Figure S12. Multi-peak fitting results of the emission spectra of A-PM-TEA under the decreasing hydrostatic pressure from 12 to 0 GPa.

Figure S13. (a) original emission spectra and (b) multi-peak fitting plots (cumulative peak of multi-peak fitting) of A-PM-TEA under the hydrostatic pressure increasing from 0 to 12 GPa.

Figure S14. Underlying characteristic peaks (1-3) resolved from the emission spectra of A-PM-TEA under the hydrostatic pressure increasing from 0 to 12 GPa.

Figure S15. Normalized and enlarged views of the underlying peaks (1-3) resolved from the emission spectra of A-PM-TEA under the hydrostatic pressure increasing from 0 to 12 GPa.

Figure S16. (a) original emission spectra and (b) multi-peak fitting plots (cumulative peak of multi-peak fitting) of A-PM-TEA under the hydrostatic pressure decreasing from 12 to 0 GPa.

Figure S17. Underlying characteristic peaks (1-3) resolved from the emission spectra of A-PM-TEA under the hydrostatic pressure decreasing from 12 to 0 GPa.

Figure S18. Normalized and enlarged views of the underlying peaks (1-3) resolved from the emission spectra of A-PM-TEA under the hydrostatic pressure decreasing from 12 to 0 GPa.

Figure S19. Intensity variations and retention rate (%) of peaks (1-3) resolved from the emission spectra of A-PM-TEA under different hydrostatic pressures.

As shown in Figure S10a-c, the *in-situ* fluorescent images and spectra of A-PM-TEA with reversible wavelength shift and intensity variation have been successfully acquired under the improved conditions with higher maximum hydrostatic pressure (~12 GPa) and more sensitive spectrometer. An apparent red shift of λ_{\max} and a sharp decline of intensity were observed (Figure S10a-b) when gradually increasing the hydrostatic pressure from 0 to 12 GPa. After withdrawing the pressure (Figure S10c), the emission λ_{\max} and intensity are completely restored to their initial state. Additionally, as seen in Figure S10e, the intensity ratio I_{627}/I_{595} rises and falls linearly when increasing and decreasing the pressure respectively. Such a linear dependence between I_{627}/I_{595} and pressure suggests that the reversible variation of emission intensity is caused by the molecular deformation and recovery under high pressure.

Given the multi-peak spectroscopic characteristics of the emission spectra in Figure S8a-b and S9a-b, the red shift of λ_{\max} in Figure S10b and the blue shift of λ_{\max} in Figure S10c might be an accumulative result of multiple characteristic emission peaks. Therefore, the multi-peak fitting analysis of the emission spectra in Figure S10b-c has been conducted to reveal the luminescence mechanism of polymaleimides under high pressures. The multi-peak fitting analysis results presented in Figure S11 and S12 suggest that there are at least three peaks (approximately 580, 620, 650 nm) beneath the emission spectra of A-PM-TEA under high pressures, which is also in

agreement with the emission characteristics in Figure S8a-b and S9a-b. The emission intensity of all these peaks (1, 2, 3) are gradually quenched with increasing the hydrostatic pressure on A-PM-TEA (Figure S14) and restored with releasing the compression (Figure S17). Meanwhile, all these emission peaks exhibit a recognizable red shift of λ_{\max} (< 15 nm, Figure S15) when rising the pressure and the λ_{\max} of peak 3 is more sensitive to the pressure and has the largest red shift. When withdrawing the pressure gradually, the opposite blue shift of λ_{\max} is observed (< 16 nm, Figure S18). It should be noted that the apparent red shift of λ_{\max} in Figure S10b and blue shift of λ_{\max} in Figure S10c are much larger than that of the resolved peaks from emission spectra in Figure S10b-c. Therefore, the apparent shift of λ_{\max} (> 30 nm) is not an overall result directly caused by the λ_{\max} shifts of three peaks under hydrostatic pressures. The retention rate plots of the emission intensity of peaks (1, 2, 3) in Figure S19 show that peak 1 is much more sensitive to the pressure than the comparable peak 2 and 3. In other words, the intensity of peak 1 declines faster than that of peak 2 and 3 under the increasing pressure. As a result, an apparent red shift of λ_{\max} (> 30 nm) originated from the competition between three characteristic peaks is observed under high pressure.

In conclusion, the multi-peak fitting analysis suggests that there are at least three underlying characteristic peaks in the emission spectra of A-PM-TEA (Figure S10b-c) under high pressure. The characteristic peaks (580, 620, 650 nm) are associated with at least three aggregation states (e.g., different packing densities) of A-PM-TEA that have different responses (especially the emission intensity) to the high pressure. As an overall result, the emission spectra of A-PM-TEA exhibit an apparent red shift λ_{\max} and a sharp decline of emission intensity when increasing the hydrostatic pressure. Because the variation of emission is induced by the reversible molecular deformation, the emission characteristics, including λ_{\max} and intensity can be fully recovered after decompression.

Second, with respect to TSCT and internal conversion

A closer intermolecular distance under high pressure is beneficial to through-space charge transfer (TSCT) and enhanced TSCT and ICT can significantly reduce the HOMO/LUMO energy gap (ΔE). According to the equation between the rate of internal conversion (k_{ic}) and the energy gap ΔE ¹,

$$K_{ic} = 10^{13} e^{-a\Delta E}$$

A smaller ΔE gives a higher k_{ic} and can intensify the non-radiative decay of the excited singlet state because of the coupling of rotational and vibrational energy-levels. The photoluminescence quantum yield (Φ) can be obtained by following functions.

$$\Phi = K_f / (K_f + K_{ic}) \propto 1 / K_{ic}$$

$$\Delta E = h\nu = hc/\lambda$$

$$\Phi \propto 10^{13} e^{ah/\lambda}$$

There is an exponential relationship between PLQY (Φ) and λ_{\max} of emission, which exactly coincides with the exponential decline plot of PLQY in Figure S6.

Generally, the emission spectra of the charge transfer complexes from TSCT and ICT are strongly influenced by the polarity of solvent². The red shift of emission spectra is always observed with increasing the polarity of the solvent. Such a red shift of λ_{\max} is also observed in Figure 3c-d (the submitted manuscript).

In summary, with a deep understanding of the emission spectra under high pressure, we realize that the statements “*that the enhanced molecular orbital overlap can promote intermolecular charge transfer*” and “*the enhanced TSCT will activate the internal conversion*” are not appropriate and inaccurate. The corrections are as follows:

1. it suggests that the reversible molecular deformation under high pressure (< 12 GPa) can significantly reduce the intermolecular distances and thus promote intermolecular trough-space charge transfer.
2. Enhanced TSCT and ICT can significantly reduce HOMO/LUMO energy gap and intensify the internal conversion.

Note: the related supplement and revision have been reorganized and highlighted in the revised version of manuscript.

Reference

1. Turro, N. J.; Ramamurthy, V.; Scaiano, J. C., *Modern Molecular Photochemistry of Organic Molecules*. University Science Books, Sausalito: 2017.
2. Lakowicz, J. R., *Principles of Fluorescence Spectroscopy*. Springer Science+Business Media, LLC: New York, 2006.

Reviewer #2 (Remarks to the Author)

Nonconventional luminophores without aromatic rings have attracted much attention and become a hot topic of research nowadays. In this paper, the authors reported the nonconventional luminescent behaviors of nonaromatic poly(maleimide)s (PMs). They achieved full-color emission (blue, green, red) of PMs via anionic polymerization of nonaromatic maleimide catalyzed by different organic bases. The resultant PMs showed aggregation-caused quenching (ACQ) instead of cluster-triggered emission (CTE) characteristic which is usually observed in most nonconventional luminophores. Finally, some simple applications were demonstrated. The results are interesting. The manuscript is recommended for publication in Nature Communications after careful revisions.

(1) The nonconventional luminescent properties of maleimide derivatives from small molecules to polymers have been reported in recent years (e.g., Chem. Commun. 2015, 51, 9733; Macromolecules 2020, 53, 3756–3764; Adv. Sci. 2021, 8, 2004299). These relevant papers may be mentioned and cited in the Introduction section.

RESPONSE:

Thanks for your kind suggestions.

As suggested, these relevant papers have been added into the introduction section. Meanwhile, the nonconventional luminescent polymers containing isolated maleimide moieties and poly(maleic anhydride)s have been summarized.

Note: the related supplement and revision have been made and highlighted in the revised version of manuscript.

(2) Because the nonconventional luminescent behaviors of PMs reported by the authors are quite different from many reported works, some theoretical calculations are needed for deep understanding of these phenomena instead of only assumptions.

Figure S1. Optimized molecular conformation and dipole moment (6 repeat units) of PMs in -C-C- and -C-N- connections calculated using uB3LYP/6-31G(d) method.

Figure S2. Excited-state HOMO/LUMO of PMs (6 repeat units) calculated using uB3LYP/6-31G(d) method.

RESPONSE:

Thanks very much for your advices.

As recommended, the optimized molecular structures and wavefunctions of PMs based on time-dependent density functional theory (TD-DFT) were calculated using Gaussian 09 with uB3LYP/6-31G(d) package. Multiwfn was utilized to obtain the electron densities, energy levels, and electronic potential distributions. VMD was used to visualize the HOMO and LUMO of PMs. The optimized molecular conformations and excited-state HOMO/LUMO of PMs are illustrated in Figure S1 and S2, respectively. In comparison, -C-N- connected PM has a larger dipole moment and a smaller HOMO/LUMO energy gap than -C-C- connected PM, which exactly coincides with their luminescence characteristics. On the basis of the theoretical calculations and the luminescence behaviors under high pressure, the understanding of the nonconventional luminescence of PMs has been improved.

(3) The authors compared the emission intensity of A-PM powders, but it is hard to make all conditions the same for different solid powders during PL tests. Thus, the absolute photoluminescence quantum yields (PLQYs) of the polymers should be given for clear comparison.

Figure S3 Emission intensity and related PLQY of A-PM powders synthesized with different Lewis bases (I, triethylamine, TEA; II, sodium bicarbonate, NaHCO₃; III, 1,8-diazabicyclo [5.4.0] undec-7-ene, DBU; IV, dimethylphenylphosphine, PMP; V, hexylamine, HA).

Table S1 PLQY of A-PM powders synthesized with different Lewis bases ($\lambda_{\text{ex}} = 365 \text{ nm}$).

No.	Lewis base	PLQY %
I	TEA	1.2
II	NaHCO ₃	1.4
III	DBU	1.5
IV	PMP	3.6
V	HA	16.2

RESPONSE:

Thanks for your suggestions.

It is indeed very difficult to compare the luminescence of powders directly by their emission intensities because the uniform test conditions are almost impossible to achieve for powders. Thus, the absolute photoluminescence quantum yields (PLQYs) of the powders have been given in Figure S3 and Table S1 for clear comparison. Fortunately, the PLQYs variations coincide with the emission intensity plot of different A-PM powders.

(4) The molecular weights and their distributions of all polymers should be given.

**Figure S4** GPC results of PMs synthesized with different Lewis bases.**Table S2.** Number-average molecular weight (Mn), weight-average molecular weight (Mw), polydispersity (Mw/Mn) of different PMs.

Sample	λ_{max} , nm	Mn	Mw	Mw/Mn
A-PM-HA	464.8	2.053×10^2 ($\pm 8.719\%$)	2.078×10^2 ($\pm 8.492\%$)	1.01
A-PM-PMP	504.0	8.578×10^2 ($\pm 1.445\%$)	1.406×10^3 ($\pm 2.681\%$)	1.64
A-PM-DBU	586.6	9.286×10^3 ($\pm 1.024\%$)	1.062×10^4 ($\pm 1.171\%$)	1.14
A-PM-NaHCO ₃	596.4	8.844×10^3 ($\pm 1.446\%$)	1.452×10^4 ($\pm 1.257\%$)	1.64
A-PM-TEA	608.6	1.024×10^4 ($\pm 0.904\%$)	1.620×10^4 ($\pm 0.801\%$)	1.58
Fr-PM-AIBN	467.4	3.825×10^4 ($\pm 0.567\%$)	4.927×10^4 ($\pm 0.251\%$)	1.29

RESPONSE:

Thanks for your suggestions.

As shown in Figure S4 and Table S2, the molecular weights and their distribution of different PMs

have been provided. These results indicate that polymerization mode and molecular weight is a major factor to determine the nonconventional luminescence of PMs. For example, a larger molecular weight of the PMs from anionic polymerization (initiated with Lewis bases) generally correlates with a longer emission wavelength (λ_{\max}). On the contrary, the molecular weight of the PM (Fr-PM-AIBN) from free radical polymerization is much larger than that of each PM (e.g., A-PM-TEA) from anionic polymerization. However, Fr-PM-AIBN only has blue emission (λ_{\max}). Such solid evidence suggests that the polymerization mode (-C-C-, -C-N- or other repeat units) should be responsible for the difference between the emission wavelengths of Fr-PM-AIBN and A-PM-TEA.

(5) For the same catalytic system, the authors should study molecular weight effect on the emission color.

Figure S5 GPC plots of PMs (A-PM-TEA) synthesized with TEA for different reaction time (60 °C, 10-360 min).

Table S3. Number-average molecular weight (Mn), weight-average molecular weight (Mw), polydispersity (Mw/Mn) of PMs synthesized with TEA for different reaction time (60 °C, 10-360 min).

Reaction time (min)	Mn	Mw	Mw/Mn	PLQY %
10	2.501×10^3	3.528×10^3	1.411	1.1 %
20	3.333×10^3	4.188×10^3	1.256	1.3 %
30	3.884×10^3	4.941×10^3	1.272	1.4 %
40	4.414×10^3	5.430×10^3	1.230	1.4 %
50	5.312×10^3	6.231×10^3	1.173	1.3 %
60	5.594×10^3	6.603×10^3	1.180	1.2 %
120	6.716×10^3	8.483×10^3	1.263	1.1 %
180	7.832×10^3	1.047×10^4	1.336	0.9 %
360	1.361×10^4	2.233×10^4	1.641	0.7 %

Figure S6 (a) *in-situ* emission spectra and (b) PLQY (%) of PMs (A-PM-TEA) synthesized with TEA for different reaction time (60 °C, 10-360 min).

RESPONSE:

Thanks for your suggestions.

Herein, the emission spectra of A-PM-TEA is employed to monitor the polymerization process and the relationship between the PLQY of the powder finally obtained and their molecular weight is established to investigate the molecular weight effect on the emission. As shown in Figure S6a, a similar result to Figure 2f in the submitted manuscript is obtained. At first stage of reaction (0-30 min), Both the emission intensity and PLQY rise with the increase of molecular weight (reaction time). Afterwards the emission intensity and PLQY continuously decline with the growth of molecular weight. Particularly, two characteristic emission bands exist in the emission spectra. The emission band of short wavelength shows a blue shift while the band of long wavelength exhibits a red shift along with the growth of molecular weight, which is remarkably similar to the concentration-dependent emission characteristics in Figure 3a-b (the submitted manuscript).

(6) *The authors mentioned TSCT effect both in “Solvent-dependent emission” and “Pressure-dependent emission”. The emission redshift and intensity decrease were both observed in “Solvent-dependent emission”, but only intensity decrease was observed in “Pressure-dependent emission”. Why did the emission wavelength keep the same with the increased pressure and TSCT effect?*

Figure S7. Pressure-dependent emission of A-PM-TEA (0-5 GPa): (a, b) *in-situ* emission spectra of A-PM-TEA under different hydrostatic pressures. Normalized emission spectra and their enlarged views of A-PM-TEA when rising (c, d) and decreasing (e, f) the hydrostatic pressures.

Figure S8. Pressure-dependent emission of A-PM-TEA (0-10 GPa): Normalized emission spectra of A-PM-TEA when rising (a) and decreasing (b) the hydrostatic pressures.

RESPONSE:

Thanks for your comments.

As noticed by reviewers, it is difficult to observe obvious changes of emission wavelengths (λ_{\max}) in Figure S7a-b (the original data in the first submitted manuscript). There are several reasons for the weak signals of emission wavelength shift.

- (1) The ratio of signal to noise (S/N) in the fluorescence spectra under high hydrostatic pressures is much lower than that in normal fluorescence spectra. In addition, emission wavelength (λ_{\max}) is generally not as sensitive as the emission intensity to pressure.
- (2) The decrease of intermolecular distance induced by a limited hydrostatic pressure (5 GPa) is insufficient to generate a significant red shift of emission λ_{\max} that can be directly observed.

In fact, a noticeable change of emission wavelengths under different hydrostatic pressures can be observed in the enlarged views of the normalized emission spectra Figure S7c-f. A slight red shift of the peaks at approximate 600, 635 and 660 nm appears and the shoulder peak at approximate 660 nm gradually rises when increasing the pressure to 5 GPa. Oppositely, a blue shift of the emission wavelengths and a fall of the shoulder peak at approximate 660 nm occur simultaneously. As shown in Figure S8, the above shift of λ_{\max} and variation of the shoulder emission around 660 nm become much more evident when increasing the maximum pressure to 10 GPa, in comparison with pressure-dependent emission of A-PM-TEA under hydrostatic pressures ranged from 0-5 GPa.

The positive results in Figure S8 encouraged us to repeat our experiment in Figure S7a-b under a large pressure range (0-12 GPa) with an advanced spectrometer. The experimental details are as follows.

2.0 g of A-PM-TEA powder was dissolved in 5.0 mL of DMF. The solvent was slowly evaporated on a hot-plate (80 °C) in order to eliminate the voids in A-PM-TEA powders and a solid sample like crystal was finally obtained.

The pressure-dependent emission of A-PM-TEA was investigated using a diamond anvil cell (DAC, Ila). The culet diameter of diamond is 500 μm . A T301 gasket (original thickness 250 μm) was preindented to 50 μm and a central hole (diameter, 200 μm) was drilled to load the solid sample of

A-PM-TEA. Ruby spheres were inserted into the central hole for in-situ pressure calibration and silicone oil was used to transmit the pressure. The photoluminescent spectra under high pressure were recorded with a confocal Raman microscope (HORIBA LabRAM HR Evolution) at room temperature. Instrument setting: ND filter = 0.1%, acq. time(s) = 0.1, accumulations = 60, grating = 600 (500 nm), excitation wavelength 532 nm and scan range 540-850 nm. The pressure was manually adjusted through a screw rod and balanced for 5 min. Then the fluorescent spectra and confocal images of A-PM-TEA under different hydrostatic pressures were acquired.

Figure S9. Pressure-dependent emission of A-PM-TEA (0-12 GPa): (a) *in-situ* fluorescent images and (b, c) emission spectra of A-PM-TEA under different hydrostatic pressures. (d, e) the emission intensities of A-PM-TEA at 594 nm and 627 nm and their ratio I_{627}/I_{594} under different hydrostatic pressures.

Multi-peak fitting analysis of the emission spectra during the pressure rise (0 to 12 GPa)

Figure S10. Multi-peak fitting results of the emission spectra of A-PM-TEA under the hydrostatic pressure increasing from 0 to 12 GPa.

Multi-peak fitting analysis of the emission spectra during the pressure fall (12 to 0 GPa)

Figure S11. Multi-peak fitting results of the emission spectra of A-PM-TEA under a decreasing hydrostatic pressure from 12 to 0 GPa.

Figure S12. (a) original emission spectra and (b) multi-peak fitting plots (cumulative peak of multi-peak fitting) of A-PM-TEA under the hydrostatic pressure increasing from 0 to 12 GPa.

Figure S13. Underlying characteristic peaks (1-3) resolved from the emission spectra of A-PM-TEA under the hydrostatic pressure increasing from 0 to 12 GPa.

Figure S14. Normalized and enlarged views of the underlying peaks (1-3) resolved from the emission spectra of A-PM-TEA under the hydrostatic pressure increasing from 0 to 12 GPa.

Figure S15. (a) original emission spectra and (b) multi-peak fitting plots (cumulative peak of multi-peak fitting) of A-PM-TEA under the hydrostatic pressure decreasing from 12 to 0 GPa.

Figure S16. Underlying characteristic peaks (1-3) resolved from the emission spectra of A-PM-TEA under the hydrostatic pressure decreasing from 12 to 0 GPa.

Figure S17. Normalized and enlarged views of the underlying peaks (1-3) resolved from the emission spectra of A-PM-TEA under the hydrostatic pressure decreasing from 12 to 0 GPa.

Figure S18. Intensity variations and retention rate (%) of peaks (1-3) resolved from the emission spectra of A-PM-TEA under different hydrostatic pressures.

As shown in Figure S9a-c, the in-situ fluorescent images and spectra of A-PM-TEA with reversible wavelength shift and intensity variation have been successfully acquired under the improved conditions with higher maximum hydrostatic pressure (~ 12 GPa) and more sensitive spectrometer. An apparent red shift of λ_{max} and a sharp decline of intensity were observed (Figure S9a-b) when gradually increasing the hydrostatic pressure from 0 to 12 GPa. After withdrawing the pressure (Figure S9c), the emission λ_{max} and intensity are completely restored to their initial state. Additionally, as seen in Figure S9e, the intensity ratio I_{627}/I_{595} rises and falls linearly when increasing and decreasing the pressure respectively. Such a linear dependence between I_{627}/I_{595} and pressure suggests that the reversible variation of emission intensity is caused by the molecular deformation and recovery under high pressure.

Given the multi-peak spectroscopic characteristics of the emission spectra in Figure S7a-b and S8a-b, the red shift of λ_{max} in Figure S9b and the blue shift of λ_{max} in Figure S9c might be an accumulative result of multiple characteristic emission peaks. Therefore, the multi-peak fitting analysis of the emission spectra in Figure S9b-c has been conducted to reveal the luminescence mechanism of polymaleimides under high pressures. The multi-peak fitting analysis results presented in Figure S10 and S11 suggest that there are at least three peaks (approximately 580, 620, 650 nm) beneath the emission spectra of A-PM-TEA under high pressures, which is also in

agreement with the emission characteristics in Figure S7a-b and S8a-b. The emission intensity of all these peaks (1, 2, 3) are gradually quenched with increasing the hydrostatic pressure on A-PM-TEA (Figure S13) and restored with releasing the compression (Figure S16). Meanwhile, all these emission peaks exhibit a recognizable red shift of λ_{max} (< 15 nm, Figure S14) when rising the pressure and the λ_{max} of peak 3 is more sensitive to the pressure and has the largest red shift. When withdrawing the pressure gradually, the opposite blue shift of λ_{max} is observed (< 16 nm, Figure S17). It should be noted that the apparent red shift of λ_{max} in Figure S9b and blue shift of λ_{max} in Figure S9c are much larger than that of the resolved peaks from emission spectra in Figure S9b-c. Therefore, the apparent shift of λ_{max} (> 30 nm) is not an overall result directly caused by the λ_{max} shifts of three peaks under hydrostatic pressures. The retention rate plots of the emission intensity of peaks (1, 2, 3) in Figure S18 show that peak 1 is much more sensitive to the pressure than the comparable peak 2 and 3. In other words, the intensity of peak 1 declines faster than that of peak 2 and 3 under the increasing pressure. As a result, an apparent red shift of λ_{max} (> 30 nm) originated from the competition between three characteristic peaks is observed under high pressure.

In conclusion, the multi-peak fitting analysis suggests that there are at least three underlying characteristic peaks in the emission spectra of A-PM-TEA (Figure S9b-c) under high pressure. The characteristic peaks (580, 620, 650 nm) are associated with at least three aggregation states (e.g., different packing densities) of A-PM-TEA that have different responses (especially the emission intensity) to the high pressure. As an overall result, the emission spectra of A-PM-TEA exhibit an apparent red shift λ_{max} and a sharp decline of emission intensity when increasing the hydrostatic pressure. Because the variation of emission is induced by the reversible molecular deformation, the emission characteristics, including λ_{max} and intensity can be fully recovered after decompression.

(7) Please explain why enhanced TSCT would facilitate internal conversions.

RESPONSE:

Thanks for your comments.

A closer intermolecular distance under high pressure is beneficial to the through-space charge transfer (TSCT) and the enhanced TSCT and ICT can significantly reduce the HOMO/LUMO energy gap (ΔE). According to the equation between the rate of internal conversion (k_{ic}) and the energy gap ΔE ¹,

$$K_{ic} = 10^{13} e^{-a\Delta E}$$

A smaller ΔE gives a higher k_{ic} and can intensify the non-radiative decay of the excited singlet state because of the coupling of rotational and vibrational energy-levels. The photoluminescence quantum yield (Φ) can be obtained by following functions.

$$\Phi = K_f / (K_f + K_{ic}) \propto 1 / K_{ic}$$

$$\Delta E = h\nu = h/\lambda$$

$$\Phi \propto 10^{13} e^{ah/\lambda}$$

There is an exponential relationship between PLQY (Φ) and λ_{max} of emission, which exactly coincides with the exponential decline plot of PLQY in Figure S3.

Generally, the emission spectra of the charge transfer complexes from TSCT and ICT are strongly influenced by the polarity of solvent ². The red shift of emission spectra is always observed with increasing the polarity of the solvent. Such a red shift of λ_{\max} is also observed in Figure 3c-d (the submitted manuscript).

(8) How did the authors confirm the concentration of the polymers in DMEM after filtered with a syringe-driven filter (0.2 μm) in “Cell viability test”? If only the original concentration was used, it would make no sense.

RESPONSE:

Thanks for your concern.

A-PM-TEA is insoluble in DMEM. Theoretically, there are no polymers in the extract solutions and the actual concentration of the extract solution is extremely difficult to be quantified. In our manuscript, only the skin-contact applications of A-PM-TEA are recommended. Therefore, the original concentration for extraction was used instead to estimate the potential cytotoxicity of A-PM-TEA, according to the Voluntary National Standards (GB/T16886.5-2017) for biocompatibility test of skin-contact medical devices and materials.

(9) The description of “Application” is too short. The Fe(III) quenching mechanism should be discussed.

Figure S19 (a) Normalized UV-Vis spectrum of Fe^{3+} and normalized excitation/emission spectra of A-PM-TEA/DMF solution. (b) UV-Vis spectra of Fe^{3+} , A-PM-TEA and A-PM-TEA + Fe^{3+} . (c)

Fluorescence decay curves of A-PM-TEA solution before and after adding Fe^{3+} . (d) Fe^{3+} -concentration dependent plots of F_0/F at different temperatures (A-PM-TEA/DMF, 3 mg/mL; $\text{Fe}^{3+}(\text{aq.})$, 0.25 mM).

RESPONSE:

Thanks for your kind suggestion.

We have strengthened the application description, especially the quenching mechanism of Fe(III) for A-PM-TEA.

As shown in Figure S19a-b, there are obvious overlaps between the absorption spectra of Fe^{3+} , A-PM-TEA+ Fe^{3+} and the excitation/emission spectra of A-PM-TEA. Therefore, the inner filter effect is inevitable for emission quenching of A-PM-TEA by adding Fe^{3+} .

In addition, the fluorescence lifetime of A-PM-TEA (Figure S19c) does not change after adding Fe^{3+} . The slope of the Fe^{3+} -concentration dependent F_0/F plots becomes smaller when rising the temperature from 25 to 50 °C (Figure S19d). These results indicate that the static quenching mechanism is involved in A-PM-TEA+ Fe^{3+} system.

Conclusively, the emission quenching of A-PM-TEA by adding Fe^{3+} should be a synergic result of the inner filter effect and the static quenching mechanism due to the formation of A-PM-TEA- Fe^{3+} complexes.

(10) Some small mistakes: Figure 2, “(d) HA and (e) TEA” should be “(e) HA and (f) TEA”; Figure 5c, the colors of emission lines for different ions were marked wrongly.

RESPONSE:

More than grateful for your kind help and correction.

We have revised these mistakes and highlighted in the revised manuscript.

Reference

1. Turro, N. J.; Ramamurthy, V.; Scaiano, J. C., *Modern Molecular Photochemistry of Organic Molecules*. University Science Books, Sausalito: 2017.
2. Lakowicz, J. R., *Principles of Fluorescence Spectroscopy*. Springer Science+Business Media, LLC: New York, 2006.

Reviewers' Comments:

Reviewer #1:

Remarks to the Author:

The authors have performed a lot of characterization to address the concerns raised by referees. Overall, the manuscript has been well revised and is suitable to be published. However, before the final decision of acceptance, the authors need to consider the following questions and give an appropriate response.

1. In my last comments, I suggested that the long-wavelength emission may come from the complex between maleimide units and the trace amount of Lewis bases. The authors exclude this possibility by the data of A-PM-NaHCO₃ and variation of TEA doping ratio. These arguments are reasonable. Then, the author concluded that the emission color of these polymers is determined by the polymerization mode and molecular weight, which mainly affect the intra-/intermolecular CT. To confirm such a conclusion, the same experiment in Figure S26 is suggested to perform for Fr-PM-AIBN with TEA mixture, and may also consider the factor of mixing time to see whether a new long-wavelength peak will appear or not.
2. In Figure 2b, the concentration of these solutions should be indicated in the caption.

Reviewer #2:

Remarks to the Author:

The authors have adequately revised their manuscript according to my previous comments and suggestions. The quality of the manuscript has been improved after the revision. I do not have further criticism of the work.

Point-by-point Responses to Reviewers

Dear Dr. Johannes Kreutzer,

Thank you for giving us the opportunity to revise our manuscript one last time. We also would like to thank the reviewers for their comments and advices to improve our paper. All these comments have been responded point-by-point. Accordingly, the manuscript and supplementary information have been carefully revised to meet the criteria for publication.

Reviewer #1 (Remarks to the Author)

The authors have performed a lot of characterization to address the concerns raised by referees. Overall, the manuscript has been well revised and is suitable to be published. However, before the final decision of acceptance, the authors need to consider the following questions and give an appropriate response.

1. In my last comments, I suggested that the long-wavelength emission may come from the complex between maleimide units and the trace amount of Lewis bases. The authors exclude this possibility by the data of A-PM-NaHCO₃ and variation of TEA doping ratio. These arguments are reasonable. Then, the author concluded that the emission color of these polymers is determined by the polymerization mode and molecular weight, which mainly affect the intra-/intermolecular CT. To confirm such a conclusion, the same experiment in Figure S26 is suggested to perform for Fr-PM-AIBN with TEA mixture, and may also consider the factor of mixing time to see whether a new long-wavelength peak will appear or not.

RESPONSE:

Sincere gratitude for your suggestion.

Figure 27 Emission of Fr-PM-AIBN with TEA. **a** Fr-PM-AIBN powders and **b** Fr-PM-AIBN/DMF solutions mixed with TEA at different concentrations (Fr-PM-AIBN:TEA, mass ratio). Concentration of Fr-PM-AIBN was 10 mg/mL.

Figure 28 Emission changes of Fr-PM-AIBN/DMF solutions over the time after adding TEA. **a** Emission spectra. **b** Emission intensity. Concentration of Fr-PM-AIBN was 10 mg/mL and mass ratio Fr-PM-AIBN:TEA = 1:1.

Figure 29 ^1H NMR and FTIR analysis of Fr-PM-AIBN before and after mixing with TEA. **a** FTIR spectra. **b** ^1H NMR spectra. Mass ratio Fr-PM-AIBN:TEA = 1:1. Fr-PM-AIBN/TEA is the sample from the Fr-PM-AIBN/TEA mixture dissolved in DMSO and recovered in ethanol. Fr-PM-AIBN/TEA (washed) is the sample from the Fr-PM-AIBN/TEA mixture dissolved in DMSO and precipitated in ethanol for 4 cycles.

Same experiment in Supplementary Figure 26 is conducted for Fr-PM-AIBN/TEA mixtures. As shown in Supplementary Figure 27a, the emission spectra of Fr-PM-AIBN powder have changed little after mixing with TEA in different mass ratios, indicating a limited influence of TEA on the emission of Fr-PM-AIBN powders. However, the emission of Fr-PM-AIBN/DMSO solutions in Supplementary Figure 27b seem sensitive to TEA and are gradually quenched by increasing the concentration of TEA in solutions. As suggested, the emission spectra of Fr-PM-AIBN/TEA solutions over the mixing time were recorded. The results in Supplementary 28a-b show a continuous decline in emission intensity after a sudden rise at the beginning. Interestingly, a new long-wavelength peak at approximately 500 nm indeed appears, which is obviously different from the result in Supplementary Figure 26. This new emission peak and the emission quenched by TEA could be ascribed to the formation of new luminescent species, the Fr-PM-AIBN/TEA complex, which can be further confirmed by FTIR and ^1H NMR analysis. FTIR in Supplementary Figure 29a

shows that the characteristics of $-\text{CH}_2-$, $-\text{CH}_3$ ($2930, 2970 \text{ cm}^{-1}$) appear in the FTIR of Fr-PM-AIBN/TEA samples in comparison with Fr-PM-AIBN. More evidently, peaks of $-\text{CH}_2-$ and $-\text{CH}_3$ (1.24, 0.84 ppm) that are different from that of the small molecular TEA (2.41, 0.92 ppm) emerges from the ^1H NMR spectra of Fr-PM-AIBN/TEA samples. These characteristic peaks are absent in the ^1H NMR of Fr-PM-AIBN and cannot be eliminated by washing the sample repeatedly. Comparatively, the minimal changes in emission of Fr-PM-AIBN powders with TEA are attribute to the slow kinetic rate of solid-phase interactions.

The emission of A-PM-AIBN/TEA suggests that the blue intrinsic emission of Fr-PM-AIBN also originates from its $-\text{C}-\text{C}-$ polymer structure, instead of the complex of Fr-PM-AIBN and residual chemicals, which generally has a longer wavelength of emission.

2. In Figure 2b, the concentration of these solutions should be indicated in the caption.

RESPONSE:

Thanks for your advice. The concentration of the solutions in Figure 2b and 2d has been provided in the caption of Figure 2.

Reviewer #2 (Remarks to the Author)

The authors have adequately revised their manuscript according to my previous comments and suggestions. The quality of the manuscript has been improved after the revision. I do not have further criticism of the work.

RESPONSE:

Thanks for your contribution to the peer review of this work.